# Polycyclic aromatics in the Chang'E 5 lunar soils

**Guangcai Zhong** [1], **Xin Yi**[1], **Shutao Gao**[1], **Shizhen Zhao** [1], **Yangzhi Mo**[1], **Lele Tian**[1], **Buqing Xu** [1], **Fu Wang**[1], **Yuhong Liao**[1], **Tengfei Li**[1], **Liangliang Wu**[1], **Yunpeng Wang** [1], **Yingjun Chen** [2], **Yue Xu**[3], **Sanyuan Zhu** [1], **Linbo Yu**[1], **Jun Li** [1], **Ping'an Peng** [1] & **Gan Zhang** [1]✉

Polycyclic aromatics are ubiquitous in the interstellar medium and meteorites, yet the search for lunar polycyclic aromatics remains a significant challenge. Here, we analyze Chang'E-5 lunar soil samples, revealing polycyclic aromatic concentrations of 5.0−9.2 μg/g (average: 7.4 ± 1.4 μg/g). Their aromatic structures are highly condensed, comparable to ~4 nm graphene sheets, and distinct from terrestrial analogs, such as wood char, soot and kerogen. While meteorite impacts are the most likely sources, the stable carbon isotope composition of polycyclic aromatics in Chang'E-5 lunar soil ($\delta^{13}$C: −5.0 ± 0.6‰ to +3.6 ± 1.3‰) is more enriched in $^{13}$C compared to that in meteorites. This enrichment suggests a de novo formation mechanism during meteorite impacts, involving the conversion of non-aromatic organic matter—which is more enriched in $\delta^{13}$C—into polycyclic aromatics. This process may play a significant role in carbon accretion in lunar regolith, as the resulting polycyclic aromatics are more stable and resistant to degradation compared to smaller organic molecules (e.g., amino acids), which are largely destroyed during impact events.

The search for organic matter on the Moon has long posed a significant challenge. Since the return of lunar samples by the Apollo missions in the 1970s, little evidence has been obtained. Early studies struggled to distinguish indigenous organic matter from terrestrial contamination in lunar samples[1,2]. At that time, methane was reported as the only indigenous organic compound definitively identified[3-5]. More recently, two non-proteinogenic amino acids, detected at concentrations below 4 ppb in Apollo 16 and 17 lunar soils, were proposed to be indigenous due to their rarity in the terrestrial biosphere and their distinct racemic mixtures[6]. Additionally, polycyclic aromatic organic matter was identified as surface coatings on three glass beads from the Apollo 17 mission using ultraviolet (UV) fluorescence imaging and Raman spectroscopy[7]. However, to the best of our knowledge, no further evidence of organic matter on the Moon has been reported.

In this study, we investigated the presence of organic matter in a lunar soil sample (CE5C0400YJFM00506) returned by the Chang'E-5 mission in 2020. The sample was collected from a loose regolith surface in the relatively flat terrain of the northeastern Oceanus Procellarum[8]. Rock fragments larger than 1 mm were removed from the regolith, leaving a highly mixed lunar soil[8]. Preliminary analyzes using optical/fluorescence microscopy, Raman spectroscopy, and pyrolysis gas chromatography-mass spectrometry (Py-GC-MS) revealed no detectable signs of organic matter (see Supplementary Information).

Here, we employ the benzene polycarboxylic acid (BPCA) method to probe polycyclic aromatics in lunar soil. Polycyclic aromatics are ubiquitous in the interstellar medium and constitute a significant fraction of organic matter in meteorites, particularly carbonaceous

[1]State Key Laboratory of Advanced Environmental Technology (SKLAET), Guangzhou Institute of Geochemistry, Chinese Academy of Sciences, Guangzhou 510640, China. [2]Shanghai Key Laboratory of Atmospheric Particle Pollution and Prevention (LAP3), Department of Environmental Science and Engineering, Fudan University, Shanghai 200438, China. [3]Laboratory of Environmental Geochemistry, Institute of Geochemistry, Chinese Academy of Sciences, Guiyang 550081, China. ✉e-mail: zhanggan@gig.ac.cn

chondrites[9,10]. Meteorite impacts are considered a key source of carbon in lunar regolith[7,11,12]. Despite this, evidence for polycyclic aromatics on the Moon has remained scarce[7]. Using the BPCA method, we identified polycyclic aromatics in the Chang'E-5 lunar soil, determined their abundance and stable carbon isotopic compositions ($\delta^{13}$C), and characterized the degree of condensation of their aromatic structures. We further discuss the fate of polycyclic aromatics on the Moon and their implications for organic carbon accretion on the Moon and early Earth.

## Results and discussion
### Abundance of polycyclic aromatics
BPCAs are benzene rings substituted with 2 to 6 carboxylic acid groups. The BPCA method involves the oxidation of polycyclic aromatics into BPCAs using nitric acid under high temperature and pressure (Fig. 1)[13]. The abundance of polycyclic aromatics is then calculated by multiplying the measured BPCA carbon content by an empirical conversion factor (see "Methods")[14,15]. It should be noted that in this study, polycyclic aromatics specifically refer to the fused benzene ring structures characterized and quantified by the BPCA method. These structures may represent either the entire molecule or only a part of it, so they are not fully described by the common terms 'polycyclic aromatic hydrocarbons,' 'polycyclic aromatic substances,' or 'polycyclic aromatic chemicals.' This method has been widely applied to studies of polycyclic aromatics in terrestrial environments[16–19].

The BPCA analyses were conducted in two triplicate batches using ~0.4 g and 1.0 g of lunar soil, resulting in a total of six analyses. BPCAs substituted with 2 to 6 carboxylic groups were detected in all lunar soil samples (Fig. 2). In laboratory blank samples, only trace amounts of B6CA were observed (Fig. S1), with concentrations less than 1% of those measured in the lunar soil. To investigate the potential presence of mineral-protected polycyclic aromatics, the residues from the first BPCA analyzes were treated with 10% hydrofluoric acid to remove silicate minerals and subsequently reanalyzed using the BPCA method.

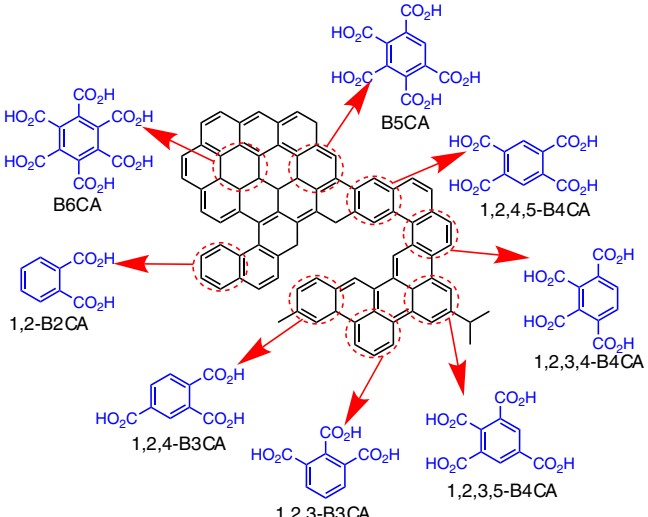

**Fig. 1 | Schematic illustration of the conversion from fused benzene rings to benzene polycarboxylic acids (BPCAs) through nitric acid oxidation.** During oxidation, a single fused benzene ring (highlighted in red) is preserved and substituted with carboxylic groups derived from adjacent carbon atoms. This process generates BPCAs with 2 to 6 carboxylic acid groups (highlighted in blue), including phthalic acid (1,2-B2CA), trimellitic and hemimellitic acids (1,2,4-B3CA and 1,2,3-B3CA), prehnitic, mellophanic and pyromellitic acids (1,2,3,4-B4CA, 1,2,3,5-B4CA and 1,2,4,5-B4CA), benzenepentacarboxylic acid (B5CA), and mellitic acid (B6CA)[13]. Additionally, di-nitrated 1,2-B2CA and mono-nitrated 1,2-B2CA, B3CA and B4CA are also produced[15].

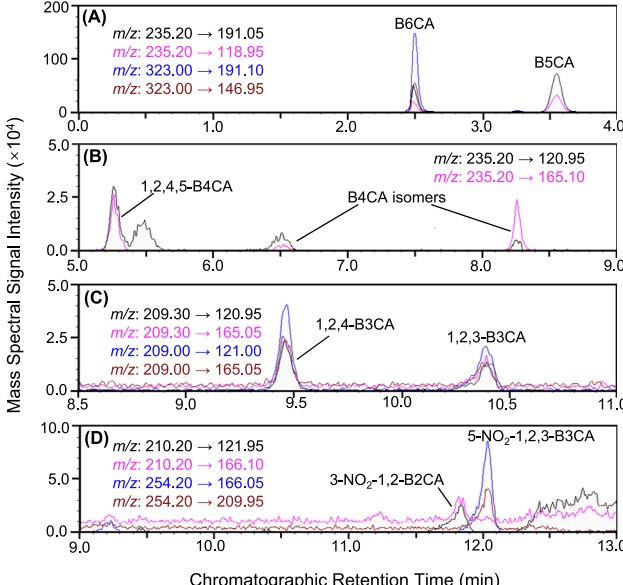

**Fig. 2 | Chromatograms of benzene polycarboxylic acids (BPCAs) molecular markers for polycyclic aromatics in Chang'E 5 lunar soil samples measured by liquid chromatography-triple quadrupole mass spectrometry (LC-MS/MS).** The LC-MS/MS ion source utilized electrospray ionization (ESI) operated in negative mode. The BPCAs were separated on a reversed-phase LC column (InfinityLab Poroshell 120 SB-C18, 4.6 × 100 mm, 2.7 μm, Agilent) using a binary gradient elution system, with 4% formic acid as mobile phase A and acetonitrile as mobile phase B. The identified BPCAs included mellitic acid (B6CA) and benzenepentacarboxylic acid (B5CA) (**A**), pyromellitic acid (1,2,4,5-B4CA) and its two isomers (**B**), trimellitic and hemimellitic acids (1,2,4-B3CA and 1,2,3-B3CA) (**C**) and 3-nitrophthalic and 5-nitro-1,2,3-benzenetricarboxylic acids (3-$NO_2$-1,2-B2CA and 5-$NO_2$-1,2,3-B3CA) (**D**).

No additional BPCAs were detected after hydrofluoric acid treatment, indicating the absence of mineral-protected polycyclic aromatics in the lunar soil.

The total carbon content in the detected BPCAs was determined to range from 0.87 to 1.62 μg C/g of lunar soil (Table 1). By multiplying the total carbon content of BPCAs by a conversion factor of 5.7[15,16], the concentrations of polycyclic aromatics in the lunar soil samples were calculated to be 5.0–9.2 μg/g (ppm) (Table 1). These polycyclic aromatics account for ~4% to 7% of the total carbon in lunar soil, based on the reported average carbon abundance of 124 ± 45 ppm in Apollo lunar soils[20].

### Degree of aromatic condensation
In high-temperature processes, the aromatization of organic matter and the condensation of aromatic rings lead to the formation of polycyclic aromatics[21–24]. For instance, heating chestnut wood chips at a temperature between 200 to 1000 °C under a nitrogen atmosphere for 5 h forms polycyclic aromatics[21]. On Earth, the primary sources of polycyclic aromatics include incomplete combustion of biomass and fossil fuel, and the thermal alteration of organic matter during burial[25–27]. The latter process produces kerogen, an insoluble macromolecular organic material with polycyclic aromatic backbones, which serves as the source of petroleum and natural gas[25]. Polycyclic aromatics are ubiquitous in terrestrial environments, with concentrations in soils and sediments spanning a wide range. For example, concentrations of 0.02–83 g/kg have been reported in a watershed affected by wildfires of varying severity[28]. Despite this variability, polycyclic aromatics generally contribute significantly to organic carbon pools. Globally, they are estimated to account for ~13% of organic carbon in the top 2 meters of soil and 4–22% of organic carbon in ocean sediments[26].

**Table 1 | Abundance of benzene polycarboxylic acids (BPCAs) molecular markers and total inferred polycyclic aromatics in the Chang'E 5 lunar soil samples**

| Lunar soil sample | Abundance (µg C/g) | | | | | | |
|---|---|---|---|---|---|---|---|
| | B6CA | B5CA | B4CAs | B3CAs | B2CAs | Total BPCAs | polycyclic aromatics |
| CE5-B1S1 | 0.73 | 0.34 | 0.02 | 0.06 | 0.03 | 1.18 | 6.73 ± 1.68 |
| CE5-B1S2 | 1.08 | 0.41 | 0.03 | 0.07 | 0.03 | 1.62 | 9.23 ± 2.31 |
| CE5-B1S3 | 0.90 | 0.34 | 0.02 | 0.04 | 0.02 | 1.33 | 7.57 ± 1.89 |
| CE5-B2S1 | 0.91 | 0.39 | 0.02 | 0.06 | 0.02 | 1.40 | 7.98 ± 2.00 |
| CE5-B2S2 | 0.92 | 0.34 | 0.02 | 0.06 | 0.01 | 1.36 | 7.73 ± 1.93 |
| CE5-B2S3 | 0.44 | 0.34 | 0.02 | 0.05 | 0.01 | 0.87 | 4.95 ± 1.24 |

B6CA and B5CA were mellitic and benzenepentacarboxylic acids, respectively. B4CAs include prehnitic, mellophanic and pyromellitic acids (1,2,3,4-B4CA, 1,2,3,5-B4CA and 1,2,4,5-B4CA); B3CAs include trimellitic, hemimellitic and 5-nitro-1,2,3-benzenetricarboxylic acids (1,2,4-B3CA and 1,2,3-B3CA and 5-NO$_2$-1,2,3-B3CA); B2CAs include phthalic and 3-nitrophthalic acids (1,2-B2CA and 3-NO$_2$-1,2-B2CA). The uncertainties in the abundance of polycyclic aromatics were calculated by applying a 25% error to the conversion factor (5.7) between total BPCA carbon and polycyclic aromatics (see "Methods")[15,16]. Triplicates from the first batch of lunar soil samples are labeled as CE5-B1S1, CE5-B1S2 and CE5-B1S3, while those from the second batch are labeled as CE5-B2S1, CE5-B2S2 and CE5-B2S3.

We compared the degree of aromatic condensation of polycyclic aromatics in the Chang'E-5 lunar soil samples with that of terrestrial analogs, including kerogens, wood char and soot, based on BPCA compositions. According to the principle of the BPCA method, more condensed polycyclic aromatic structures produce a higher proportion of highly substituted BPCA homologs (refer to Fig. 1). These terrestrial analogs correspond to the primary sources of polycyclic aromatics in terrestrial environments. Among these analogs, we analyzed natural thermal series of kerogen (Fig. 3A, #1–6 and #7–19), artificial thermal series of kerogen (Fig. 3B, #1–6 and #7–12) and wood char (Fig. 3B, #13–18). The natural thermal series of kerogen vary in thermal maturity, which is related to their formation temperature[25]. Similarly, the artificial thermal series of kerogen and wood char vary in their treatment temperatures. A general increase in highly substituted BPCAs (i.e., B6CA and B5CA) is observed from low to high thermal maturity or from low to high treatment temperatures in these series. This trend suggests that temperature is a key factor controlling the degree of condensation of polycyclic aromatics[21,26].

The BPCA compositions reveal that the polycyclic aromatics in the lunar soil samples are generally more condensed than those in natural kerogen samples, except for those from Mesoproterozoic carbonatite strata (Fig. 3A, #5–6). In the lunar soil, B6CA and B5CA predominate, accounting for 96% ± 0.7% of the total BPCAs. Although the B6CA content in kerogen samples from Mesoproterozoic carbonatite strata is higher than in the lunar soil, these kerogen samples also contain significant amounts of B4CA and B3CA (~10%), indicating the presence of both highly condensed and smaller polycyclic aromatic clusters. Compared to artificial thermal series of kerogen and wood char treated at ≤600 °C (Fig. 3B, #1–14), the polycyclic aromatics in the lunar soil are more condensed. However, they are less condensed than wood char produced at ≥800 °C, coal coke pyrolyzed at ~1000 °C, and soot derived from fossil fuel combustion (Fig. 3B, #16–23). These findings suggest that the polycyclic aromatics in the lunar soil likely formed at temperatures above 600 °C. However, an upper limit for their formation temperature cannot be definitively established below 800 °C, as fragmentation of polycyclic aromatic structures may occur, as discussed below.

Polycyclic aromatics in terrestrial soils and sediments are generally less condensed than those in the lunar soil, even in samples from locations exposed to intensive biomass burning or fossil fuel combustion. Results from the BPCA method indicate that B6CA and B5CA account for no more than 80% of the total BPCAs in forest soils following recent wildfires, paddy field soils with a long history of rice straw burning, and urban air particles[15,29,30]. A significant fraction of polycyclic aromatics released into terrestrial soils and sediments are

formed at relatively lower temperatures. For example, most natural fires reach temperatures of ~500 °C[31].

Currently, there are no BPCA data available for other extraterrestrial samples for comparison. To the best of our knowledge, the Murchison meteorite is the only extraterrestrial sample that has been analyzed for BPCA molecular markers[32]. In this case, B6CA and B5CA together accounted for ~65% of the total BPCAs. However, the analytical procedures used, which involved refluxing the samples with nitric acid for 27 h, were significantly different from ours, making direct comparison unreliable. Insoluble organic matter (IOM), often referred to as kerogen-like or macromolecular organic matter, typically constitutes more than 75% of the organic matter in carbonaceous chondrite meteorites[9]. A proposed molecular structure model for the IOM in the Murchison meteorite consists of 2–4 fused benzene rings[33], but graphene sheets have also been identified in this meteorite[34]. Polycyclic aromatic hydrocarbons (PAHs) and fullerenes (C60 and C70) have been detected in the Allende meteorite at concentration levels of ppm and ppb, respectively[35]. Additionally, the insoluble carbon in the Tagish Lake carbonaceous chondrite meteorite exhibits an exclusively aromatic character and contains fullerenes[36]. In the recently returned samples from the carbonaceous asteroid (162173) Ryugu, several hollow and solid organic nanoglobules, detected by X-ray absorption imaging, were confirmed to be either aromatic or highly aromatic[37]. The H/C ratio is closely related to the degree of condensation of polycyclic aromatics[21]. A study of IOM from 75 chondrites revealed a wide range of H/C ratios (8.4 to 95)[9], suggesting that the condensation degree of polycyclic aromatics in meteorites may vary significantly. This variability warrants further investigation using the BPCA method.

We roughly estimate the size of the aromatic domains of polycyclic aromatics in the lunar soil based on the BPCA compositions. First, small PAH molecules or their derivatives can be excluded as dominant contributors, as they produce negligible amounts of fully substituted mellitic acid (B6CA). The absence of detectable organic matter released during Py-GC-MS analysis, even when heating the samples to 800 °C under a helium flow, further supports the lack of small PAH molecules and suggests that the polycyclic aromatics exist in a refractory form under these conditions. Fullerenes (C60 and C70) and carbon nanotubes (CNTs) can also be ruled out, as their oxidation primarily yields fully substituted B6CA with minimal amounts of less substituted BPCAs[38]. Instead, graphene-like structures, which are refractory and consistent with the observed BPCA compositions, may serve as a suitable surrogate for the polycyclic aromatics in the lunar soil. Theoretically, a graphene sheet composed of a 10 × 10 benzene-ring array would yield a BPCA composition similar to that observed in the lunar soil (Fig. 3 and Table S7). This graphene sheet is estimated to have a unit size of ~4.12 nm in diameter (Fig. S2), based on a carbon-

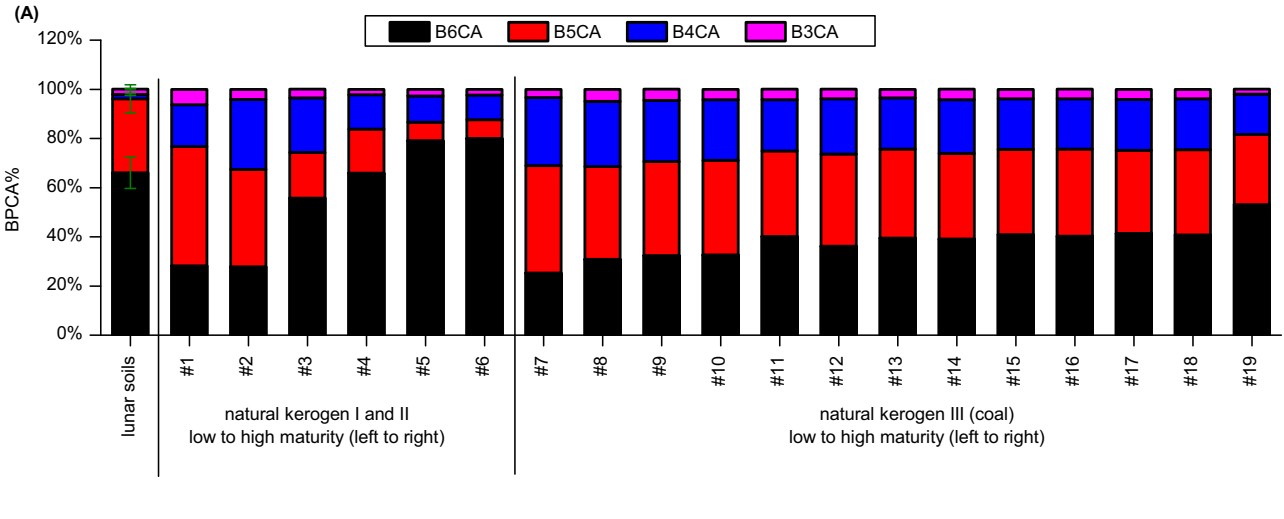

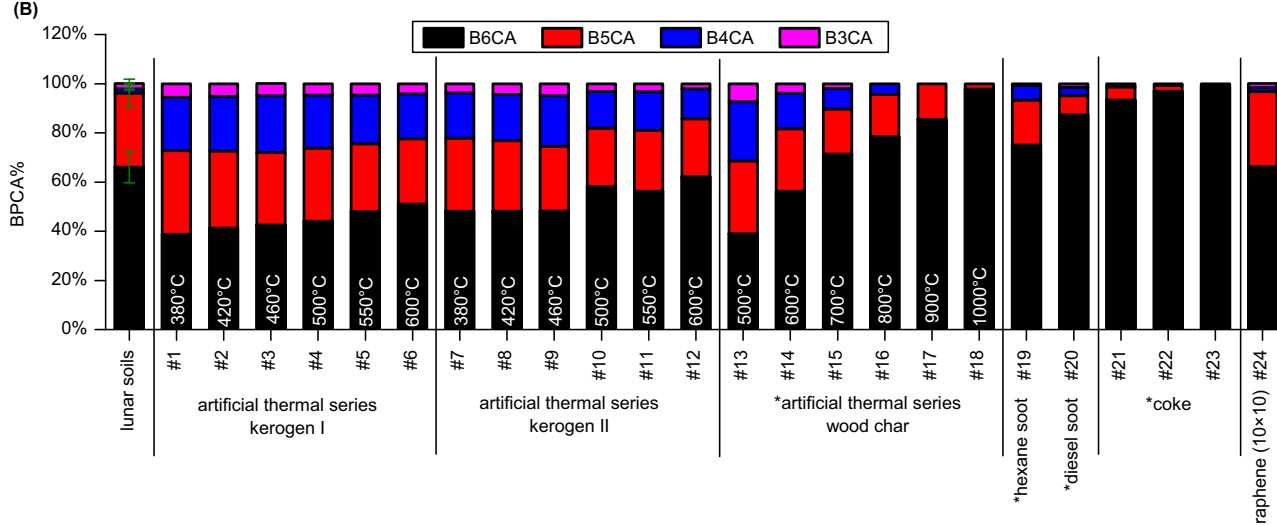

**Fig. 3 | Benzene polycarboxylic acids (BPCAs) compositions of polycyclic aromatics in the Chang'E 5 lunar soils and terrestrial analogs (i.e., kerogen, wood char and soot).** The results for natural kerogen are presented in (**A**), while those for the artificial thermal series of kerogen and wood char, as well as soot, are shown in (**B**). The y-axis represents the percentage of individual BPCAs relative to the total BPCAs in carbon content. The BPCAs substituted with two carboxylic groups (B2CAs) were excluded from the calculation, consistent with previous studies, to enable direct comparison[21,74]. Errors for samples other than the lunar soils are not shown, as they are less than 2%. The "*" in the labels on the x-axis denotes data from the literatures[21,74], while other results are from this study (see Supplementary Methods, Table S4–6). The result of graphene sheet composed of a 10 × 10 benzene-ring array (#24) is based on theoretical calculations (see Supplementary Method, Table S7). The maturity of natural type I and II kerogen was determined using vitrinite reflectance (Table S4). The maturity of natural type III kerogen was determined based on volatile matter content on a dry ash-free basis (Table S5).

carbon bond length of 0.142 nm. Experimental validation of the BPCA composition for such graphene is impractical due to the lack of appropriate standards. Among terrestrial analogs, the wood char produced at 700 °C exhibits the BPCA composition most similar to that of the lunar soil (Fig. 3B, #15). A quantitative relationship model has been established between the size of aromatic structures and the H/C ratio for such materials[39]. For the wood char produced at 700 °C, the model predicts an aromatic cluster size of a 13 × 13 benzene-ring array, based on its H/C ratio. This model slightly overestimates the size of aromatic clusters as revealed by high-resolution transmission electron microscope (HRTEM) observations[40]. These comparisons support the estimate of the size of the aromatic domains of polycyclic aromatics in the lunar soil. A concurrent study also observed few-layer graphene in the Chang'E-5 lunar soils by HRTEM, although its abundance was not determined, and terrestrial contamination has yet to be ruled out[41]. Notably, the polycyclic aromatics in our sample are significantly more condensed than those detected on Apollo 17 glass beads, which were estimated to consist of only 5 benzene rings[7].

## Fate of polycyclic aromatics on the Moon

The most probable events responsible for the polycyclic aromatics in the Chang'E-5 lunar soils are meteorite impacts. Carbon accretion through meteorite impacts is estimated to be sufficient to explain the carbon abundance detected in the lunar surface regolith[7]. Typically, more than 75% of meteoritic organic matter consists of IOM, which is rich in polycyclic aromatics[9]. It is also known that fragments of carbonaceous meteorites can survive accretion[42]. The Yutu-2 rover of the Chang'E-4 mission detected glassy material on the Moon, with a high concentration (47%) of carbonaceous chondrites in a two-meter-sized crater that formed less than one million years ago[43]. Carbonaceous chondrite fragments have also been identified in Apollo samples[42]. This supports a scenario in which meteorite impact remnants serve as a direct source of polycyclic aromatics.

However, the stable carbon isotope compositions (δ13C) of the polycyclic aromatics in the lunar soil favor a de novo formation mechanism during meteorite impact events. Shocks may induce physico-chemical transformation in amorphous carbon to form

**Table 2 | Stable carbon isotope compositions (δ13C) of benzene polycarboxylic acids (BPCAs) molecular markers for the polycyclic aromatics in the Chang'E 5 lunar soils**

| Lunar soil sample | δ13C (‰, VPDB) | |
| --- | --- | --- |
| | B6CA | B5CA |
| CE5-B1S1 | −0.34 ± 0.65 | 2.07 ± 0.18 |
| CE5-B1S2 | −4.99 ± 0.56 | 3.55 ± 1.29 |
| CE5-B1S3 | 0.19 ± 0.56 | 3.12 ± 1.29 |

The δ13C values are expressed relative to Vienna Peedee Belemnite (VPDB). B6CA and B5CA were mellitic and benzenepentacarboxylic acids, respectively. Only the two most abundant BPCAs (B6CA and B5CA) of the first batch of lunar soil samples (CE5-B1S1, CE5-B1S2 and CE5-B1S3) were analyzed for stable carbon isotope compositions.

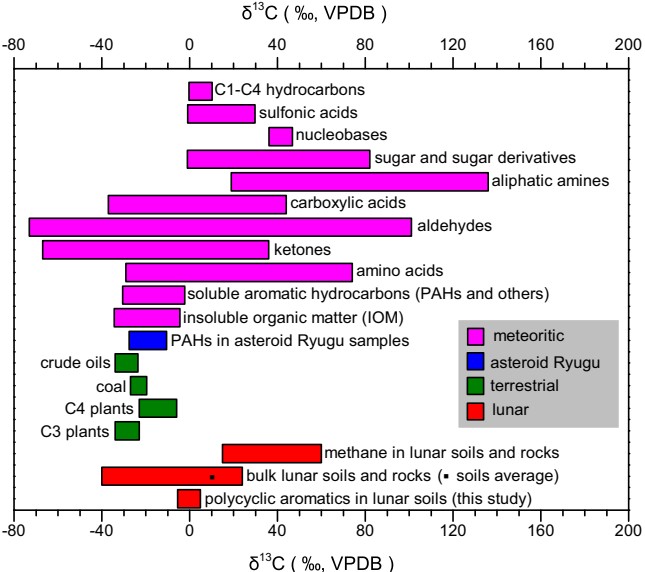

**Fig. 4 | Stable carbon isotope compositions of the polycyclic aromatics in the lunar soils, and comparison with those of terrestrial organic materials and other extraterrestrial samples.** The δ13C values are expressed relative to Vienna Peedee Belemnite (VPDB). The literature values of chondritic meteorites are compiled for insoluble organic matter[9,66,79], soluble aromatic hydrocarbons[36,49–54], amino acids[67,80–85], ketones[67,68], aldehydes[67,68], carboxylic acids[36,69,71,86–88], aliphatic amines[70], sugars and sugar derivatives[66,89], nucleobases[87], sulfonic acids[90] and C1–C4 hydrocarbons[88]. The literature values for bulk lunar soils and rocks[12], methane in lunar soils and rocks[4,5], polycyclic aromatic hydrocarbons (PAHs) in asteroid Ryugu samples[54], and terrestrial organic materials (C3 plants, C4 plants, coal and crude oils)[91] are also compiled.

complex structures, such as carbon nanotubes, graphene and fullerenes[44], which could eventually be redeposited on the regolith after temporally short impact events[7]. We used preparative high-performance liquid chromatography (Prep-LC) to purify and isolate the two most abundant BPCAs, i.e., B5CA and B6CA (see "Methods"), derived from nitric acid oxidation of the first batch of lunar soil samples. Their compound-specific δ13C values were then measured using liquid chromatography-isotope ratio mass spectrometry (LC-IRMS) (see "Methods"). The δ13C values of B6CA range from −4.99‰ to +0.19‰, while those of B5CA range from +2.07‰ to +3.55‰ (Table 2). These δ13C values of BPCAs can be regarded as the δ13C values of their polycyclic aromatic precursors[45]. First, these values are distinct from those of polycyclic aromatics on Earth (-63‰ to -15‰)[46–48], indicating an extraterrestrial origin.

We further compared these values with those of extraterrestrial organics from the literature (Fig. 4). A large dataset of δ13C values for

IOM in chondritic meteorites, which is rich in polycyclic aromatics, ranges from approximately -34‰ to -4.5‰[9]. Similarly, the δ13C values of soluble aromatic hydrocarbons in asteroid Ryugu and chondritic meteorites (approximately −31‰ to −2.2‰) are mostly more depleted than those of the polycyclic aromatics in the Chang'E-5 lunar soils[36,49–54]. In contrast, soluble non-aromatic organic matter in chondritic meteorites is frequently more enriched in 13C (Fig. 4). Thus, the polycyclic aromatics in the Chang'E-5 lunar soil samples likely originated from the transformation of bulk organic matter, which could include IOM, soluble aromatic compounds, and non-aromatic organic matter from meteoroids, asteroids, and comets that impacted the Moon. These de novo formation processes would produce polycyclic aromatics with δ13C values reflecting a mixture of these three types of organic matter. Additionally, pyrolysis of methane, which is implanted into the lunar regolith by solar wind and occurs at concentrations up to ppm levels[4,5], could also be induced by meteorite impacts to form polycyclic aromatics[55]. This pathway could contribute polycyclic aromatics enriched in 13C, given the relatively high δ13C values of methane in the lunar regolith (+15‰ to +60‰)[4,5].

Even when isotopic fractionation is taken into account, the above inference remains valid. Isotopic fractionation of IOM during meteorite impacts is expected to be insignificant. For instance, pyrolysis of IOM from the Allende meteorite at 250–800 °C does not alter its δ13C values[56], while similar pyrolysis of the Murchison meteorite (250–937 °C) results in only a minor decrease of less than 1.7‰[56,57]. The decrease is due to loss of some 13C-enriched parts weekly attached to the main network of IOM[57]. The transformation of methane and soluble aromatic and non-aromatic organic matter is expected to yield polycyclic aromatics with δ13C values that are either similar or more enriched. This is because, under complete transformation, no significant isotopic fractionation occurs. However, if carbon is partially lost through evaporation during impact, the lighter 12C is more likely to evaporate preferentially, leaving the heavier 13C enriched in the resulting polycyclic aromatics. This process is analogous to isotopic fractionation observed during combustion. In the combustion of fossil fuels, for example, the majority of carbon is lost as CO2, leaving residual particles with δ13C enrichments of no more than 4‰ for solid and liquid fossil fuels and no more than 11‰ for natural gas[58].

Constraining the contributions of different isotopic pools to the δ13C values of polycyclic aromatics in lunar soil is challenging due to the limited data on the mass percentages and δ13C values of bulk soluble organic matter in meteorites. Additionally, the extent of carbon loss during the transformation of soluble organic matter and methane into polycyclic aromatics during impacts remains unclear. We provide an estimation only based on data from the relatively well-studied Murchison meteorite. In this meteorite, IOM constitutes ~75% of the organic matter, with a δ13C value of −15.5‰, while soluble organic matter accounts for the remaining 25%[57,59]. The δ13C value of the soluble organic matter is assumed to be +6‰, based on measured values of +5 to +7‰ for organic matter of benzene-methanol, methanol, and hot water extracts of this meteorite[59]. In a simplified scenario where all carbon from the meteorite's organic matter is transformed into highly condensed polycyclic aromatics without isotopic fractionation, the resulting polycyclic aromatics would have a δ13C value of −10.1‰, as calculated by mass balance. This value is still lower than the δ13C range observed for polycyclic aromatics in lunar soil (−5.0 to +3.6‰). To reconcile this discrepancy, lunar soil methane (with δ13C values of +15 to +60‰) would need to contribute at least 7% of carbon (at +60‰) to reach the lower end of the δ13C range (−5.0‰) for lunar polycyclic aromatics, and up to 55% of carbon (at +15‰) to reach the upper end (+3.6‰).

The aromatic structures of polycyclic aromatics in lunar soil exhibit distinct characteristics compared to their terrestrial counterparts. To elucidate these differences, we employ two key indicators: (1) the percentage of B6CA + B5CA in the total BPCAs ($P_{B6CA+B5CA}$), which

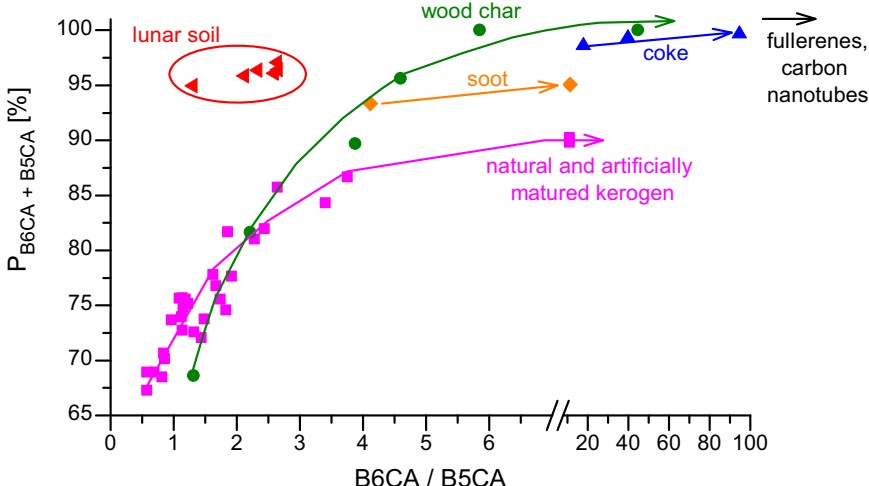

**Fig. 5 | The co-evolution of overall condensation degree and the confined aromatic domains of polycyclic aromatics.** The percentage of mellitic acid (B6CA) and benzenepentacarboxylic acid (B5CA) in the total benzene poly-carboxylic acids ($P_{B6CA+B5CA}$) serves as an indicator of the overall condensation degree of the polycyclic aromatics, while the B6CA/B5CA ratio reflects the size of the confined aromatic domains. The data presented in this figure are derived from a reorganization of the data in Fig. 3. The arrowed lines depict the co-evolution trajectory of $P_{B6CA+B5CA}$ and B6CA/B5CA for terrestrial analogs.

reflects the overall condensation degree of the polycyclic aromatics, and (2) the B6CA/B5CA ratio, which serves as an indicator of the size (e.g., graphene sheets) and/or dimensionality (e.g., carbon nanotubes, nano-onions, fullerenes) of the confined aromatic domains. Our analysis reveals that the polycyclic aromatics in lunar soil do not align with the co-evolution trajectory of $P_{B6CA+B5CA}$ and B6CA/B5CA observed in terrestrial analogs (Fig. 5), which is primarily governed by temperature, as previously discussed. Instead, the lunar soil samples occupy the upper-left region of the diagram, deviating from the thermal evolution pathways of kerogen/coal, coal coke, wood char, and flame soot. This deviation suggests the influence of additional processes, such as shock-wave fragmentation during impact events and/or post-depositional solar irradiation-induced fragmentation of large gra-phene structures[22,60,61]. These processes likely result in the formation of small, graphene-like sheet structures dispersed throughout the lunar soil. The fragmentation should not result in isotopic fractionation, as it does not involve carbon loss or transformation into non-polycyclic aromatic structures.

## Implications to organic carbon accretion on the Moon and early Earth

We propose that graphene-like polycyclic aromatics may be dis-tributed across the lunar surface and accumulate in regions that experienced intensive meteorite impacts throughout the Moon's geological history. The transformation of labile organic matter (e.g., amino acids) into highly condensed polycyclic aromatics, which are more stable and more likely to persist under lunar surface and irra-diation conditions, could facilitate the accretion of extra-lunar organic carbon on the Moon. The samples analyzed in this study were returned by the Chang'E-5 mission from the mare plain in the northeastern Oceanus Procellarum, a region characterized by younger basalt ages (~2.0 Ga) compared to most lunar basaltic magmatism, which ceased around 2.9–2.8 Ga[62]. This implies that the mare plain has a shorter history of meteorite impacts relative to other lunar regions. Therefore, we recommend future analyzes of polycyclic aromatics in additional lunar soil samples from diverse regions using the BPCA method. Such studies would help assess the regional distribution of polycyclic aro-matics on the lunar surface and establish connections between their abundance and regional crater-counting data. This approach would provide deeper insights into the fate and preservation of polycyclic aromatics on the Moon[62,63].

Meteorite impacts were estimated to contribute ~400 ppm car-bon to lunar soil[12]; however, analyses of Apollo lunar soils revealed an average carbon content of only $124 \pm 45$ ppm[20]. This discrepancy suggests that a significant portion of carbon may have been lost through evaporation during meteorite impacts on the Moon. The average velocity of meteoroids impacting the Moon ranges from 13 to 18 km/s[64]. Simulations indicate that a porous impactor with a diameter of 15 cm, striking the lunar surface at 15 km/s, can generate shock pressures exceeding 30 GPa[43]. Additionally, lunar impact flash mon-itoring over the past two decades has recorded temperatures ranging from 1500 to 5800 K[65]. Meteorite impacts are more destructive to organic matter in the absence of a buffering atmosphere. In contrast to the frequent detection of diverse organic matter in meteorites on Earth[53,66–70], conclusive evidence of organic matter on the Moon has rarely been reported in previous studies.

We conducted targeted analyzes of ~80 soluble organic com-pounds in the Chang'E-5 lunar soil samples, including amino acids, amines, monocarboxylic acids, aldehydes, and ketones (see Supple-mentary Information). These compounds are commonly investigated in extraterrestrial samples[6,68,70–72]. Methylamine and ethylamine were detected in the Chang'E-5 samples at concentrations of 15–18 ppb and 2.0–2.6 ppb, respectively (Table S8–12). However, further analyzes, such as compound-specific isotope analysis, are required to rule out potential contamination. Most other compounds were undetectable, with a few present at levels only slightly above the method detection limit or detectable in some but not all triplicates. These results suggest that small organic molecules may have been largely erased during meteorite impacts on the Moon. Our findings have implications for the search for organic carbon and biomolecules on other planetary bodies, such as Mars and the early Earth, where similar processes may have occurred.

## Methods
### Calculation of polycyclic aromatics abundance in BPCA method
The abundance of polycyclic aromatics was estimated by multiplying the measured BPCA carbon content by an empirical conversion factor[14,15]. This factor was previously determined using polycyclic aromatic standards and chemicals encompassing a wide range of molecular structures, from small to large condensed aromatic systems, including polycyclic aromatic hydrocarbons, fullerenes, carbon nano-tubes, hexane soot, and carbon lamp black[14,15]. The conversion factor

represents an average value for all these chemicals, with an estimated uncertainty of 25%, corresponding to the relative standard deviation of conversion factors derived from individual chemicals[14,15]. Graphite falls outside the analytical window of this method due to its resistance to nitric acid oxidation[73].

BPCAs are highly specific markers for polycyclic aromatic compounds. In our previous study, we compiled data from our work and the literature on BPCA yields from non-polycyclic aromatic organic matter, including 24 species[16]. BPCAs were undetectable in most species, and when detectable, the yields were very low (≤9.1 mg/g) compared to those for polycyclic aromatic compounds (175 ± 21 mg/g, based on carbon content). Additionally, the BPCA compositions of these non-polycyclic aromatic species were distinct from those observed in lunar soil in this study[16]. Furthermore, the potential presence of polycyclic aromatic impurities in these non-polycyclic aromatic materials could not be ruled out[16], which is particularly concerning given their very low BPCA yields.

### Nitric acid oxidation of lunar soils

We analyzed BPCA molecular markers in the lunar soil sample CE5C0400YJFM00506. The analysis was conducted in two batches. The first batch involved direct triplicate analysis of -0.4 g of the lunar soil for BPCA molecular markers. The second batch included triplicate analysis of about 1.0 g of the lunar soil following preliminary extraction with hot water. The hot water extracts were used to analyze soluble organic compounds, including amino acids, amines, aldehydes, ketones, and monocarboxylic acids, while the residues were subsequently analyzed for BPCA molecular markers.

For the first batch, each of the three aliquots (-0.4 g) of the lunar soil sample was weighed into a 10-mL glass ampoule using a stainless-steel spoon. Distilled nitric acid (2 mL, -68%) was added to the ampoule using a glass pipette. The ampoule mouth was covered with aluminum foil to prevent contamination by airborne particles before being sealed with a gas flame. The sealed ampoule was placed in a stainless-steel reaction vessel with a 100-mL polytetrafluoroethylene (PTFE) liner. The vessel was heated in an oven at 180 °C for 8 h. To prevent ampoule explosions due to heating, 100 μL of water was added to the PTFE liner to balance the vapor pressure inside and outside the ampoules. After nitric acid oxidation, the mixture was transferred into a 4-mL vial using a glass pipette. Each sample was centrifuged at 3500 rpm ($2700 \times g$) for 5 min, and the supernatant was transferred into a 100-mL flask. The residue was rinsed four times with 1 mL of distilled ultrapure water, centrifuged, and the supernatants were combined in the 100-mL flask. It was confirmed that four rinses were sufficient to extract all BPCAs from the residue.

For the second batch, each of the three aliquots (-1.0 g) of the lunar soil sample was weighed into a 10-mL glass ampoule using a stainless-steel spoon. Distilled ultrapure water (2 mL) was added to the ampoule using a glass pipette. The ampoule was sealed, placed in a stainless-steel reaction vessel, and heated in an oven at 100 °C for 24 h. The solution and insoluble residue were transferred into a 4-mL glass vial and centrifuged at 3500 rpm ($2700 \times g$) for 5 min. The supernatant was transferred to a 10-mL glass ampoule for future analysis of water-soluble compounds, while the residue was dried at 90 °C under a nitrogen stream. The dried residue was evenly divided into two 10-mL glass ampoules, and the nitric acid oxidation process was repeated as described for the first batch. After oxidation, the solutions from the two sub-samples were recombined for analysis.

To investigate the presence of mineral-protected polycyclic aromatics, the solid residue remaining after nitric acid oxidation was treated with hydrofluoric acid (HF) and reanalyzed for BPCA molecular markers. The residue was transferred into a 25-mL perfluoroalkoxy (PFA) centrifuge tube and rinsed three times with 10 mL of 10% hydrofluoric acid, followed by two rinses with 10 mL of ultrapure water. After each rinse, the supernatant was discarded following

centrifugation at 3500 rpm ($2700 \times g$) for 5 min. The final solid residue was dried at 90 °C under a nitrogen stream and analyzed for BPCA molecular markers using the same procedures as described above.

Three procedural blank samples were included for each type of analysis. Quartz powder (200–250 mesh) was used as the matrix for the blanks in the preliminary analysis of BPCA molecular markers in lunar soils. Basalt powder (100 mesh) was used as the matrix for the blanks in the analysis of BPCA molecular markers in the lunar soil residues after HF treatment.

### Purification of BPCA extracts

To prepare the BPCA extracts for liquid chromatography-triple quadrupole mass spectrometry (LC-MS/MS) analysis, it was necessary to remove salts and concentrate the extracts. The BPCA extract in the 100-mL flask was dried using a rotary evaporator with a water bath temperature of 70 °C. The flask was rinsed twice with 1 mL of distilled ultrapure water, and the rinses were combined. The combined solution was then subjected to cation removal using a glass column (inner diameter: 1.2 cm, length: 30 cm) packed with cation exchange resin (Dowex 50 WX8 400, Sigma Aldrich).

Prior to use, the column was activated by sequentially eluting it with one column volume of ultrapure water, one column volume of 2 M NaOH, two column volumes of ultrapure water, and one column volume of 2 M HCl. The resin was further rinsed with ultrapure water until the conductivity of the eluent fell below 2 μS cm⁻¹. Finally, the column was eluted with 100 mL of distilled ultrapure water to ensure complete activation.

The BPCA extract was loaded onto the activated column and eluted with 10 mL of distilled ultrapure water. The eluent was collected in a 100-mL flask and dried using a rotary evaporator with a water bath temperature of 70 °C. The dried sample was redissolved in 2 mL of distilled ultrapure water and purified again using the cation exchange resin column. The eluent was collected in a 100-mL flask and dried using a rotary evaporator. The sample in the flask was rinsed twice with 1 mL of ultrapure water, and both rinses were transferred to the same 2-mL glass vial. The sample in the vial was dried at 90 °C under a nitrogen stream and then redissolved in 250 μL of distilled ultrapure water for subsequent analysis.

### Measurement by liquid chromatography-triple quadrupole mass spectrometry

The samples purified using the cation exchange resin column and concentrated to 250 μL were analyzed for BPCAs using LC-MS/MS, a highly sensitive method for BPCA measurement. The LC-MS/MS instrument (LCMS-8050, Shimadzu) was equipped with an InfinityLab Poroshell 120 SB-C18 column (4.6 × 100 mm, 2.7 μm, Agilent). Mobile phase A consisted of 4% formic acid in ultrapure water, while mobile phase B was acetonitrile. The total flow rate of the mobile phases was set to 0.4 mL/min. The mobile phase gradient was programmed as follows: mobile phase A was maintained at 98% from 0.0 to 3.0 min, decreased to 80% from 3.0 to 9.9 min, further decreased to 5% from 9.9 to 10.0 min, held at 5% from 10.0 to 13.0 min, increased to 98% from 13.0 to 13.1 min, and finally maintained at 98% from 13.1 to 15.0 min. The column oven temperature was set to 30 °C, and the sample plate temperature was maintained at 15 °C. The injection volume was 10 μL.

The ion source of the LC-MS/MS utilized electrospray ionization (ESI) operated in negative mode. The MS/MS parameters were configured as follows: interface temperature, 300 °C; block temperature, 400 °C; desolvation line temperature, 250 °C; nebulizing gas (nitrogen) flow rate, 3 L/min; heating gas (zero air) flow rate, 10 L/min; and drying gas (nitrogen) flow rate, 10 L/min. Argon was used as the collision gas at a pressure of 270 kPa. The mass spectrometer was operated in multiple reaction monitoring (MRM) mode, with the MRM parameters detailed in Table S2.

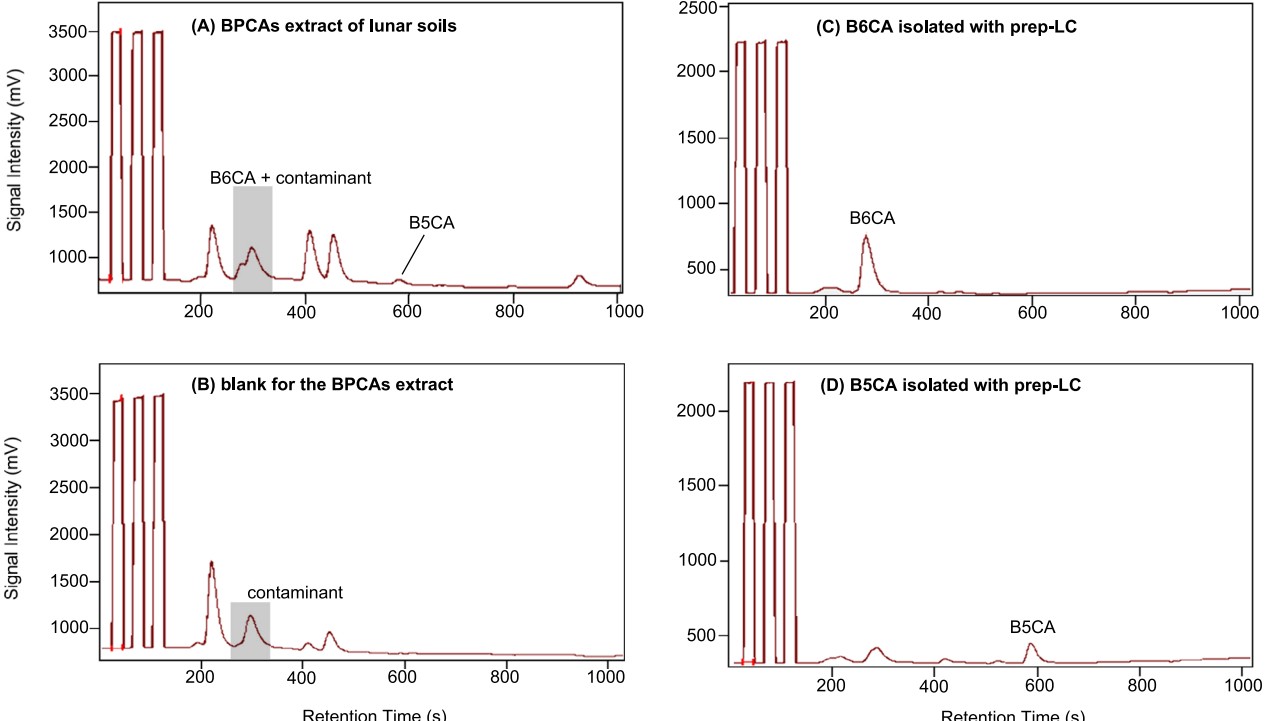

**Fig. 6 | Representative chromatograms of the benzene polycarboxylic acids (BPCAs) derived from Chang'E 5 lunar soil samples measured by liquid chromatography-isotope ratio mass spectrometry (LC-IRMS).** The chromatograms (mass-to-charge ratio: 44) are for direct LC-IRMS measurement of BPCAs extracts (**A**) and its corresponding blank sample (**B**); as well as for LC-IRMS measurement of B6CA and B5CA isolated from the BPCAs extracts of lunar soil samples with preparative liquid chromatography (prep-LC) (**C, D**).

Nine-point calibration curves ($r^2 \geq 0.99$) were established for the quantification of each BPCA, using BPCA standards at concentrations of 0.01, 0.02, 0.04, 0.08, 0.1, 0.4, 0.8, 1.0, and 4 ng/μL. The BPCA standards included B6CA, B5CA, 1,2,4,5-B4CA, 1,2,4-B3CA, 1,2,3-B3CA, 3-NO$_2$-1,2-B2CA, 5-NO$_2$-1,2,3-B3CA, and 1,2-B2CA. Standards for two B4CA isomers (1,2,3,4- and 1,2,3,5-B4CA) were commercially unavailable; therefore, their quantification was based on the calibration curve of 1,2,4,5-B4CA[13,14,16,74]. Other nitrated BPCAs were not detected due to the unavailability of their commercial standards.

BPCAs were not detectable in the blank samples (signal-to-noise ratio < 3), except for B6CA (Fig. S1). However, the B6CA levels in the blanks contributed ≤1% of the B6CA measured in the lunar soil extracts. The method detection limit for BPCA molecular markers in the lunar soils was defined as the average blank value plus three times its coefficient of variation ($n = 3$). All target BPCAs were detectable in the BPCA extracts of the lunar soils, except for 1,2-B2CA (Fig. S1).

**Measurement of stable carbon isotope compositions of BPCAs**
Stable carbon isotope compositions ($\delta^{13}$C) were measured for the most abundant BPCAs (B6CA and B5CA), but not for the other BPCAs due to their low concentrations. After the first batch of BPCA extracts were analyzed by LC-MS/MS, they were further processed for $\delta^{13}$C analysis of B6CA and B5CA. For $\delta^{13}$C measurements, B6CA and B5CA were preliminarily isolated using preparative liquid chromatography (prep-LC). The high-purity B6CA and B5CA obtained from this process were then analyzed for $\delta^{13}$C using liquid chromatography-isotope ratio mass spectrometry (LC-IRMS).

The LC-IRMS system comprised an UltiMate 3000 HPLC system coupled to a Delta V Plus IRMS via an Isolink interface (Thermo Scientific)[75]. Sample separation was achieved using a reversed-phase column (Poroshell 120 Phenyl Hexyl, 4.6 × 150 mm, 2.7 μm, Agilent) with gradient elution of binary aqueous mobile phases. Mobile phase A

consisted of phosphoric acid (pH = 1), while mobile phase B was a sodium phosphate buffer (pH = 6). Organic compounds eluted from the column were continuously oxidized into CO$_2$ using a mixed solution of phosphoric acid and sodium persulfate. The resulting CO$_2$ was introduced into the IRMS for $\delta^{13}$C measurement.

Preliminary isolation using prep-LC was necessary because direct measurement of B6CA in the lunar soil samples by LC-IRMS revealed an interfering peak (Fig. 6A). This interfering peak was also observed in the chromatograms of the blank samples (Fig. 6B). The contaminant, likely of terrestrial origin ($\delta^{13}$C = -36.1 ± 1.3 ‰), could not be resolved despite efforts to improve chromatographic separation. These efforts included increasing the acidity of mobile phase A (up to three times the original concentration) and slowing the gradient elution rate of mobile phase A.

To address this issue, we employed a preliminary isolation of B6CA and B5CA using preparative liquid chromatography (prep-LC; LC-20AT/SPD-M20A, Shimadzu). Separation was achieved using a reversed-phase column (Poroshell 120 SB-C18, 4.6 × 150 mm, 2.7 μm, Agilent) with an aqueous mobile phase A (phosphoric acid, pH = 1.4) and an organic mobile phase B (acetonitrile)[27]. The use of an organic mobile phase generally enhances separation efficiency in reversed-phase chromatography.

For each analysis, one injection of the BPCA extracts previously measured by LC-MS/MS was introduced into the prep-LC, delivering ~50 ng C of B6CA and 30 ng C of B5CA. Excellent chromatographic separation of both B6CA and B5CA was achieved (Fig. 7). The eluted B6CA and B5CA were collected using a fraction collector (FRC-10A, Shimadzu). The organic mobile phase in the collected fractions was completely removed by nitrogen evaporation at 90 °C for 1 hour, ensuring compatibility with subsequent $\delta^{13}$C measurements using LC-IRMS. After evaporation, the samples were redissolved in 50 μL of ultrapure water. The prep-LC method demonstrated high recoveries for B6CA and B5CA (≥85%)[27].

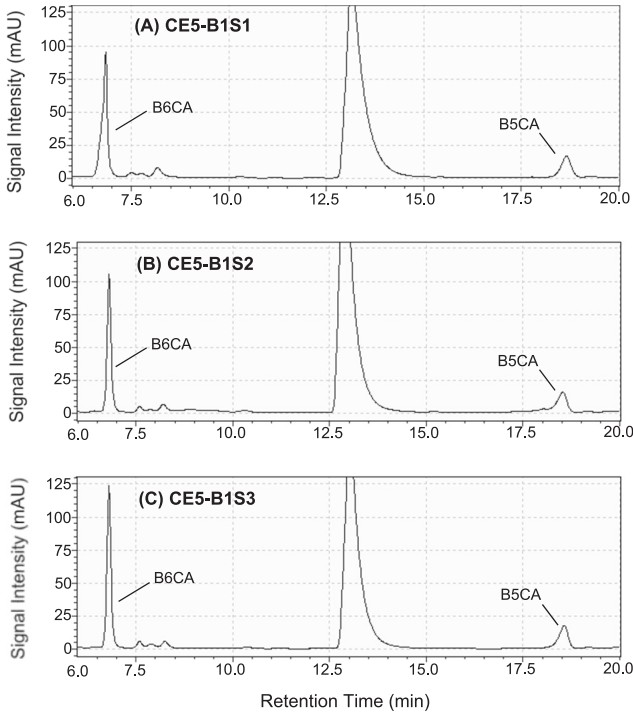

**Fig. 7 | Chromatograms of the benzene polycarboxylic acids (BPCAs) extracts of the Chang'E 5 lunar soil samples measured by preparative liquid chromatography (prep-LC).** The prep-LC was equipped with a photo-diode array (PDA) detector. The three chromatograms (wavelength: 240 nm) are for the first batch of Chang'E 5 lunar soil samples, namely CE5-B1S1 (**A**), CE5-B1S2 (**B**) and CE5-B1S3 (**C**).

Although B6CA and B5CA were preliminarily isolated using prep-LC, an analytical LC column was still employed during subsequent LC-IRMS measurements. This step was necessary to separate any potential contaminants introduced during the prep-LC procedures from B6CA and B5CA. The LC-IRMS method described earlier was used for $\delta^{13}C$ measurements of the purified B6CA and B5CA fractions, with an injection volume of 50 μL. After purification, the peak that partially coeluted with B6CA in direct LC-IRMS measurements was no longer present (Fig. 6, A vs. C), and the $\delta^{13}C$ values of B6CA were significantly more enriched (approximately −15‰ to−30‰ compared to −5‰ to +0.2‰; Table S3).

To correct the $\delta^{13}C$ values, B6CA and B5CA derived from nitric acid oxidation of maize char were isolated using prep-LC and measured for $\delta^{13}C$ using the same method. Maize char, produced through incomplete combustion or pyrolysis of biomass, is commonly used as a reference material for $\delta^{13}C$ analysis of BPCAs due to its high enrichment in polycyclic aromatics[27,45]. The $\delta^{13}C$ value of the bulk char is assumed to represent the reference $\delta^{13}C$ value for the polycyclic aromatics or the BPCAs derived from their nitric acid oxidation[27,45]. The maize char used in this study had a relatively enriched $\delta^{13}C$ value (−13.0 ± 0.18‰), as determined by elemental analyzer-IRMS. An offset, calculated from the difference between the measured and reference $\delta^{13}C$ values for B6CA (or B5CA) in the maize char, was applied to correct the $\delta^{13}C$ values measured for B6CA (or B5CA) in the lunar soil samples (Table S3).

The BPCA extract of maize char was prepared by nitric acid oxidation of 5 mg of maize char, followed by drying using the same procedures as those applied to the lunar soil samples. The dried extract was redissolved in 10 mL of ultrapure water, yielding solutions with B6CA and B5CA concentrations of ~10 ng/μL and 16 ng/μL (in carbon content), respectively. To ensure accurate correction, B6CA (or B5CA) in the maize char was isolated using prep-LC in amounts similar to those used for the lunar soil samples. Prior to prep-LC injection, the

BPCA extracts of the maize char were mixed with a procedural blank of the BPCA extracts from the lunar soils. The volume of the procedural blank used for mixing matched the volume of the BPCA extracts from the lunar soil samples injected for prep-LC separation.

### Analysis of BPCA molecular markers for coals and kerogen

The kerogen samples were extracted from rocks by removing minerals using hydrochloric acid and hydrofluoric acid, followed by the elimination of soluble organic matter using a solvent mixture of benzene, acetone, and methanol[76]. The thermal maturity of the kerogen samples was determined using vitrinite reflectance[25]. The rocks used for extracting type I and II kerogen were collected from the following locations: the Alum Formation shale in northwestern Europe (#1, Table S4), the Xiamaling Formation in Tianjin, China (#2, Table S4), the Cambrian Maidiping Formation in Sichuan, China (#3–4, Table S4), and the Ediacaran Dengying Formation in Sichuan, China (#5–6, Table S4).

The coal samples were obtained from local markets in China. Their thermal maturity was assessed based on the volatile matter content on a dry ash-free basis, which ranged from 6.1% to 48.3%[77], covering a wide spectrum of maturity levels (Table S5). Pyrolyzed kerogen samples (Table S6) were prepared using a high-pressure, semi-closed pyrolysis system[78].

The BPCA molecular markers of the coal and kerogen samples were measured with a well-established liquid chromatography method for terrestrial samples rich in organic matter[16]. The liquid chromatography was equipped with a photodiode array detector (LC-20AT/SPD-M20A, Shimadzu) and a reversed-phase column (Poroshell 120 SB-C18, 4.6 × 100 mm, 2.7 μm, Agilent). Separation was achieved using with an aqueous mobile phase A (phosphoric acid, pH = 1.4) and an organic mobile phase B (acetonitrile). The mobile phase was delivered at a total flow rate of 0.4 mL/min, with the column oven maintained at 30 °C, and the injection volume set to 10 μL. BPCA identification was achieved by comparing both retention times and UV-Vis absorption spectra (190–400 nm). BPCA quantification was conducted based on peak areas measured at 240 nm using external calibration curves.

## Data availability

Source data are provided in tables of the main text and Supplementary Information.

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

## Acknowledgements

We extend our gratitude to all the staff of China's Chang'E-5 project for their dedication and efforts in successfully returning lunar samples. The samples analyzed in this study were allocated by the China National Space Administration (CNSA). We thank Yigang Xu from the Guangzhou Institute of Geochemistry, Chinese Academy of Sciences, for facilitating sample acquisition. We are also grateful to Sasho Gligorovski from the

Guangzhou Institute of Geochemistry, Chinese Academy of Sciences, and Simon George from Macquarie University, Australia, for their constructive comments and editorial assistance with the manuscript. This work was supported by the National Natural Science Foundation of China (Grants 42192511 and 42030715) and the Key Research Program of the Chinese Academy of Sciences (Grant ZDBS-SSW-JSC007-8).

## Author contributions

G.Zhang designed the project. G.Zhong, X.Y., Y.M., S.G., L.T., F.W., T.L., L.Y., S.Zhu performed the sample analysis. G.Zhong collected analytical data. G.Zhang and G.Zhong wrote the draft manuscript. L.W. and Y.W. provided the kerogen samples. Y.C. provided the coal samples. All authors including J.L., S.Z., P.P., B.X., Y.L. and Y.X. reviewed and edited the manuscript.

## Competing interests

We declare no competing interests.
