## [Transparent Peer Review file · Nature Communications]

Polycyclic aromatics in the Chang'E 5 lunar soils

Corresponding Author: Professor Gan Zhang

Version 0:

Reviewer comments:

Reviewer #1

(Remarks to the Author)

This paper reports the first detection of graphene like PAH structures in Chang'E 5 mission return samples from the far side of the moon. Direct solvent extraction and pyrolysis GCMS revealed no PAHs but after nitric acid oxidation, large PAH graphene like structures with an estimate 4 to 6 nm diameter in size are decomposed into highly substituted benzoic acids (BPCAs). These were subsequently analyzed by HPLCMS/MS for concentrations and HPLC-IRMS for isotopic ratios of two most abundant BPCAs (with 6 and 5 carboxyl substitutions respectively). $\delta^{13}\text{C}$ of polycyclic aromatics ranges from $-5.0\pm 0.6\%$ to $+3.6\pm 1.3\%$, inferring a de novo formation of polycyclic aromatics during carbonaceous meteorite impacts, which involves a conversion of non-aromatic organic matter into polycyclic aromatics.

The results of this study are novel and important for understanding organics in extraterrestrial bodies. I enthusiastically support the publication of this paper in Nature communications, after revisions carefully addressing the following issues:

- The reported concentration of Chang'E samples of BPCAs is around 7.4 ug/g, which is high given the nature of the moon. It is important for this paper to compare and list the concentrations of BPCAs previously reported in other extraterrestrial and terrestrial samples (including the experimental samples from this study), and discuss the reasons for various concentration ranges.
- How many aromatic rings are expected, if the graphene sheet is around 4 to 5 nm in diameter, and with the corresponding BPCA distributions? Can authors use a molecular simulation software, with the constraints of BPCA distributions (e.g., B6CA/B5CA, and others), to generate an image of a model graphene sheet for the lunar soil samples?
- Have experimental impact studies using hyper velocity guns generated similar graphene structures from organic matter? Have previous studies found organics from impact craters containing more graphene like structures than non-impact regions on extraterrestrial samples?
- Fig.2 chromatogram from LCMSMS is quite busy, not always easy to read for less technical persons. It is the central data for this paper, so it is better to do it well. Although it is ok to use this figure, I like to see some kind of histogram plot to see the relative distributions of all BPCAs. The elution order of various BPCAs on chromatogram is less important than showing the relative concentrations of different compounds. If the concentration contrast is too large, you can use log scale for the Y axis to reduce the contrast. I really like to see a histogram comparison of all BPCA distributions in various samples mentioned in this study, including the experimental terrestrial samples, heated coal etc, as well as those previously reported in extraterrestrial samples such as Ryugu and Murchison etc. Although B6CA/B5CA ratio and percentages are very informative, other smaller BPCAs may also be informative of structures. A histogram including all compounds will help illustrate the differences more thoroughly. I see data for Chang'E samples are listed in Table 1. But table listing is much less appealing than a histogram figure, and less easy for a visual comparison with other samples.
- Fig.3A and B are a bit confusing. I see two Y axis on the left and right sides of the plot (percent B6CA+B5CA, B6Ca/B5CA ratiion respectively), but it takes huge efforts to read the caption to figure out what is what. The figure needs to be more intuitive for comprehension.
- Have there any published BPCA distributions of graphene sheet? In particular, if the 4-6 nm, 10 by 10 to 12 by 12 graphene sheet would generate similar BPCA distributions (e.g., B6CA to B5CA ratio), the data would be a welcome support for the paper. Line 155: Theoretically, a graphene sheet with a 10×10 benzene-ring array would give a B6CA/B5CA ratio of 2.18, and a 12×12 array would give a ratio of 2.73. This is theoretical. The actual BPCA distribution would depend on the efficiency of nitric acid oxidation (e.g, using the conditions reported in this paper). Has anyone conducted experiment to show this?
- If the Mare plain region has a shorter impact history than other regions on the moon, the chances of completely eliminating other organic matter and transforming them into graphene would be lower. This contradicts with the points made in the paper

somewhat. It is very surprising that PYGCMS revealed no organics and PAHs at all. It is hard to imagine impacts have been so fully covered the whole region and left no original organics. Impacts cannot possibly eliminate all compounds, unless overlapping impacts have occurred to cover every inch of the lunar soil, at least in the sampling region? Some explanations are needed here.

- Line 275, what is the concentration of distilled nitric acid?

- Line 247. Asteroids (and carbonaceous chondrites falling onto Earth, like Murchison) also do not atmosphere and are subjected to impacts. They contain lots of soluble organic compounds and organics did not get converted into graphene.

- Table 2, what is the bulk $\delta^{13}\text{C}$ value of the lunar soil?

- Line 280: "after addition of 100 μL water into the PTFE liner to prevent explosion of the ampoule". What does this mean?

Reviewer #2

(Remarks to the Author)

Background Info

Zhong et al., have undertaken a comprehensive analysis of polycyclic aromatic hydrocarbons (PAHs) in the Chang'E 5 lunar soil samples for the first time. They detected highly condensed graphene sheet-like PAHs and demonstrated that they are unlike those found in terrestrial settings. Using isotopic evidence they indicate that the PAHs could have formed from bulk meteoritic organic matter (both soluble/free and insoluble/macromolecular organic matter). The authors then suggest that this may mean that smaller (soluble organic molecules), such as amino acids may thus have been destroyed during impact events on the surface of the moon. Additionally, the authors suggest that similar effects could lead to the destruction of small organic molecules on other planetary surfaces, with minimal atmospheres or no atmosphere at all.

Major Comments

Figure 3a and 3b – Are the error bars included for the kerogen data? If so, are they just smaller than the white data point symbol?

Figure 3a and 3b – The graphs are somewhat confusing what does the colour bar show and what does the white dot indicate? This needs to be described in the figure caption.

Figure 3c – I suggest you to put data for insoluble organic matter isolated from meteorites. It is difficult to understand how similar or dissimilar the graphene-like PAHs you have reported are to the major component in carbonaceous chondrite meteorites.

Furthermore, it would be better to also discuss in text more clearly about the differences between the PAHs within the lunar soil samples and those in meteorites or the aromatic rich portions that compose the insoluble organic matter in meteorites. There is a study from quite some time ago that uses nitric acid to break down the insoluble organic matter within the Murchison CM2 carbonaceous chondrite ([https://doi.org/10.1016/0016-7037\(77\)90076-X](https://doi.org/10.1016/0016-7037(77)90076-X)). I suggest you to look at it and see if there is any way to compare their data to yours. Another paper discusses the general structure of meteoritic insoluble organic matter and might be useful as well (doi: 10.1111/j.1945-5100.2010.01122.x).

Table 1 – The caption states that a 25% uncertainty was used, why? Please explain this.

Did you try to search for small organic molecules? From what I can understand you haven't searched for these molecules, but you suggest they might have been erased. I suggest you to look for amino acids or carboxylic acids, which are relatively abundant in meteorites. There are a number of methods that have been utilised recently to detect amino acids in the Ryugu samples for very small sample sizes (~1-2 mg to 10's or 100's mg of sample, see <https://doi.org/10.3390/life13071448>).

Line 218-222 – The authors state that methane from solar wind can contribute to the solar regolith. It would be useful to explain whether a combination of solar wind derived methane and insoluble organic matter in meteorites or other carbon bearing phases in meteorites (e.g. carbonate) could generate the isotopic signatures of the graphene-like PAHs. Indeed, previous work simulating impacts has shown that small molecules in the interior portions of meteorites can survive impact events (e.g. DOI: 10.1089/ast.2008.0327).

Furthermore, it has also been shown that ^{13}C and D can be preferentially released from insoluble organic matter in meteorites when simulating impact events. This is because the portions of the insoluble organic matter that contain these heavy isotopes are likely bound by weaker bonds to the structure of the insoluble organic matter (see DOI: 10.1111/j.1945-5100.2007.tb00238.x). Therefore, it is possible that some isotopic fractionation between the meteoritic organic matter and the end product would have occurred. As such, I think you need to show that the lunar soils are devoid of small organic molecules like amino acids and carboxylic acids to give weight to you proposed theory of formation for the graphene-like PAHs.

Overall, the article, once revised, could appropriate for publication in Nature Communications, but it would require additional analysis. The authors should utilise sensitive techniques to search for soluble organic compounds, such as those commonly found in meteorites (e.g. amino acids and carboxylic acids). This will enable them to tell if the graphene-like PAHs formed from bulk meteorite organic matter or from another source, such as the interaction of insoluble meteoritic organic matter and solar wind CH_4 that was implanted into the lunar surface.

Minor Comments

Please change all instances of polycyclic aromatics to polycyclic aromatic hydrocarbons (including in the title), as this is the correct term and also polycyclic aromatics isn't strictly grammatically correct.

The English should be improved throughout, as there are many cases where words are missing (e.g. the) or used incorrectly (e.g. is instead of are). While I was able to follow the article for the most part, there were a number of sentences that I had to re-read several times and if the overall English was improved, I think it would greatly aid in the communication of the authors scientific findings.

Version 1:

Reviewer comments:

Reviewer #1

(Remarks to the Author)

The manuscript has been improved in the first round of revision. However, I would like to see following issues more carefully addressed:

Line 18-19: Please provide a clear explanation why the particular isotope range indicates de novo formation.

Fig.1 legend: The model structure shown is a polycyclic aromatic hydrocarbon or PAH. But the paper kept using "polycyclic aromatics" throughout the paper. If the purpose is to include aromatic structures containing oxygen, nitrogen heteroatoms etc., this should be clarified. How would the presence of oxygen or nitrogen affect the nitric acid oxidation products?

Fig.2 legend: Key HPLC parameters, like mobile phase and HPLC column condition should better be stated in the figure caption.

Line 106: High temperature processes under vacuum and absence of oxygen. If oxygen is present, the organics will be converted into CO₂ in high temperature. Also, clarify how high the temperature needs to be to produce PAHs.

Fig.3 legend. Again, people often say polycyclic aromatic hydrocarbons or PAHs. What is the reason of saying polycyclic aromatics in this paper? If there is a special reason for using polycyclic aromatics rather than PAHs, the reason should be explained clearly.

Line 240: Need more clear reasoning here why and what isotope fractionation and processes are involved, so that the carbon isotope values of B5CA and B6CA indicate de novo formation processes. For example, what percentage mixing of IOM and soluble compounds with different isotopic values would lead to observed values.

Fig.4 legend: Apart from mixing different isotope pools, what other isotope fractionation processes may be involved in the de novo synthesis?

line 252: For the observed carbon isotope values of B5CA and B6CA, try to make some estimates of different sources: how much of the solar wind implanted methane with $\delta^{13}C$ of +15 to 60 permil methane may have contributed to the formation of lunar organics? what isotope fractionation processes may be involved?

Line 272: What are the possible isotope effect of these processes?

Line 302: Decomposition and evaporation?

Reviewer #2

(Remarks to the Author)

Thank you very much for revising your manuscript. After the revisions that you have made I believe there is sufficient evidence to support your conclusions that small organic molecules within impacting meteorites could be destroyed and form the highly aromatic graphene sheet-like organic matter present in the lunar soil samples returned by the Chang E 5 mission. Therefore, I am happy for the article to be accepted in Nature communications.

Dear Reviewers,

We sincerely appreciate the time and effort you have dedicated to reviewing our manuscript. Your insightful comments have been invaluable in enhancing the quality and clarity of our work.

This resubmission comes four months after the initial review, as additional analyses of soluble organic compounds in the lunar soils were necessary to further support our conclusions.

While a major revision was required for our previous manuscript, we have made substantial revisions in response to your comments. These revisions were aimed at improving the readability of the manuscript and providing stronger evidence to support our conclusions. However, the key findings and conclusions of our study remain unchanged.

We are deeply grateful for the opportunity to have our manuscript reviewed again and appreciate your continued feedback.

Faithfully yours,

Gan Zhang

Professor of Organic Geochemistry

Guangzhou Institute of Geochemistry

Chinese Academy of Sciences

Answers to Reviewer Comments

Notes:

- **Black text** indicates the reviewers' comments.
- **Blue text** represents the authors' responses to the reviewers' comments.
- **Yellow highlighting** in the track-changed manuscript marks the revisions made.
- The notes labeled “**Line-x**” refer to the line numbers in the revised manuscript, corresponding to the specific corrections and responses to the comments.

Reviewer #1 (Remarks to the Author):

This paper reports the first detection of graphene like PAH structures in Chang'E 5 mission return samples from the far side of the moon. Direct solvent extraction and pyrolysis GCMS revealed no PAHs but after nitric acid oxidation, large PAH graphene like structures with an estimate 4 to 6 nm diameter in size are decomposed into highly substituted benzoic acids (BPCAs). These were subsequently analyzed by HPLCMS/MS for concentrations and HPLC-IRMS for isotopic ratios of two most abundant BPCAs (with 6 and 5 carboxyl substitutions respectively). $\delta^{13}\text{C}$ of polycyclic aromatics ranges from $-5.0\pm 0.6\%$ to $+3.6\pm 1.3\%$, inferring a de novo formation of polycyclic aromatics during carbonaceous meteorite impacts, which involves a conversion of non-aromatic organic matter into polycyclic aromatics.

The results of this study are novel and important for understanding organics in extraterrestrial bodies. I enthusiastically support the publication of this paper in Nature communications, after revisions carefully addressing the following issues:

- The reported concentration of Chang'E samples of BPCAs is around 7.4 ug/g, which is high given the nature of the moon.

Answer:

Thanks for your comment. Yes, the polycyclic aromatics concentration levels were high given the nature of the moon, thus we strongly believe that it is an important finding. We need to note that “7.4 $\mu\text{g/g}$ ” represents the average concentration of polycyclic aromatics, not

BPCAs, which are molecular markers used in our study to quantify polycyclic aromatics. The concentrations of polycyclic aromatics are calculated based on the measured BPCA concentrations.

Previous studies have reported an average carbon concentration of $124 \pm 45 \mu\text{g/g}$ in lunar soils (Line-97). In our study, the average concentration of polycyclic aromatics in Chang'E-5 lunar soils is $7.4 \pm 1.4 \mu\text{g/g}$, indicating that they are a significant contributor to lunar soil carbon.

This finding is not unprecedented. Meteorite impacts, particularly those involving carbonaceous chondrites, have been proposed as a major source of carbon in lunar soils. The carbon in carbonaceous chondrites is primarily derived from insoluble organic matter (IOM), also referred to as kerogen-like or macromolecular organic matter in some studies. The structural backbone of IOM consists of polycyclic aromatic systems. We have added this information to the revised manuscript (Please refer to Line-54, Line-210 to 214, and Line-300 to 302).

A related question is why polycyclic aromatics were not commonly detected in previous studies if their concentrations are relatively high in lunar soils. Our findings provide insight into this issue. We discovered that the polycyclic aromatics in lunar soils are graphene-like, which may explain why they were overlooked in earlier studies. For example:

- Gas chromatography (GC) is sensitive to small polycyclic aromatic molecules (2–7 rings) but not to non-volatile, graphene-like structures.
- Graphene-like structures are refractory, meaning they do not produce volatiles or breakdown products detectable by pyrolysis-GC-MS (Py-GC-MS).
- Spectroscopic techniques, such as optical/fluorescence microscopy and Raman spectroscopy, are rarely capable of detecting polycyclic aromatic structures at the 1 ppm ($\mu\text{g/g}$) level but depends greatly on a ‘lucky strike’ on the region(s) of interest (RIO). In our study, we also employed these techniques but did not find any RIO (please refer to Line-48 to 50).

Our successful detection of polycyclic aromatics is attributed to the use of a novel analytical tool: the BPCA method. This method has a broad analytical window, capable of detecting a wide range of polycyclic aromatic structures, from small molecules to fullerenes and carbon nanotubes (Line-328 to 330). The BPCA method is highly sensitive, enabling the detection of

polycyclic aromatics at the 1 ppm level with ease.

It is important for this paper to compare and list the concentrations of BPCAs previously reported in other extraterrestrial and terrestrial samples (including the experimental samples from this study), and discuss the reasons for various concentration ranges.

Answer:

[About Extraterrestrial Samples]

We appreciate the suggestion to compare BPCA levels in lunar soils with those in extraterrestrial samples. However, such data are currently unavailable. Reviewer #2 referenced a study on BPCA analysis of the Murchison meteorite, published in 1977. To our knowledge, this is the only study reporting BPCA analysis of extraterrestrial samples. Unfortunately, the BPCA method used in that study differs significantly from ours, making direct comparisons impractical. Our BPCA method has been widely used for analyzing polycyclic aromatics in terrestrial samples and, to our knowledge, is the first to be applied to extraterrestrial samples. We have added a discussion on this topic in the revised manuscript (please refer to Line-163 to 168).

[About Terrestrial Environmental Samples]

BPCA concentrations in terrestrial samples can vary widely depending on their organic carbon content. A more meaningful comparison would focus on the contribution of polycyclic aromatics to the total organic matter in terrestrial soils and sediments. Indeed, polycyclic aromatics are a significant component of organic matter in these terrestrial samples. We have included a discussion on this point in the revised manuscript (Line-106 to 117 and Line-156 to 162).

[About Experimental Samples]

Similar to the condensation degree, the concentrations of polycyclic aromatics in experimental samples are temperature-dependent. Aromatization is enhanced at higher temperatures, leading to a wide range of concentrations, from 0 to nearly 1 g/g (i.e., 100%). Due to this variability, compiling such data is not practical.

- How many aromatic rings are expected, if the graphene sheet is around 4 to 5 nm in diameter,

and with the corresponding BPCA distributions? Can authors use a molecular simulation software, with the constraints of BPCA distributions (e.g, B6CA/B5CA, and others), to generate an image of a model graphene sheet for the lunar soil samples?

Answer:

We appreciate the suggestion to use molecular simulation software. We currently lack the expertise or collaborative resources to perform such simulations. Instead, we have provided theoretical calculations of BPCA compositions for graphene sheets ranging from 5×5 to 15×5 benzene arrays using mathematical equations. These calculations are detailed in the **Supplementary Method** section of the revised manuscript. Additionally, we have included the BPCA compositions of these graphene sheets in **Fig. 3B** (#24) and **Table S7** of the revised manuscript. Notably, the BPCA composition of a 10×10 graphene sheet closely matches that of the lunar soils, as demonstrated in the figure and table.

- Have experimental impact studies using hyper velocity guns generated similar graphene structures from organic matter?

Answer:

Graphene sheet structures can indeed form through the heating of organic matter at sufficiently high temperatures. For example, heating rice straw in a nitrogen atmosphere at 700°C produces char, which is rich in graphene-like structures. Similarly, graphene sheet structures are expected to form under the high temperatures generated during meteorite impacts. We have added this information to the revised manuscript (please refer to Line-106 to 111 and Line-197 to 201). Thus, while hypervelocity is not a prerequisite, high temperatures are essential for the formation of graphene-like structures.

Additionally, we found that the BPCA compositions of the lunar soils are distinct from those of terrestrial analogs, suggesting a unique formation mechanism for polycyclic aromatics in lunar soils. We propose a mechanism involving the fragmentation of larger graphene structures due to shock waves during meteorite impacts and have provided references to support this conclusion. Please refer to Line-271 to 275 in the revised manuscript for further details.

Have previous studies found organics from impact craters containing more graphene like

structures than non-impact regions on extraterrestrial samples?

Answer:

This is indeed an interesting topic, but we have been unable to find any related report in the literature. To our knowledge, there are no quantitative results on graphene-like structures in lunar samples. If the comparison is extended to impact craters on Earth, it may not be practical due to the potential conversion of mostly large amount terrestrial organic matter into graphene-like structures during high-temperature impact events. Additionally, other terrestrial sources, such as biomass and fossil fuel combustion, could mask the polycyclic aromatics formed during impact events (Line-106 to 117 and Line-156 to 162).

- Fig.2 chromatogram from LCMSMS is quite busy, not always easy to read for less technical persons. It is the central data for this paper, so it is better to do it well. Although it is ok to use this figure, I like to see some kind of histogram plot to see the relative distributions of all BPCAs. The elution order of various BPCAs on chromatogram is less important than showing the relative concentrations of different compounds. If the concentration contrast is too large, you can use log scale for the Y axis to reduce the contrast.

Answer:

Thanks for your kind concern. Detecting carbon components in lunar soil is indeed a challenging task, with false positives being a major concern. Figure 2 demonstrates excellent chromatographic separation, clear peaks, and well-defined peak shapes for our target compounds. This provides strong evidence that BPCA molecular markers are detectable in the Chang'E-5 lunar soils. Therefore, we believe it is valuable to retain this figure in the main text, and we are proud of the beautiful original chromatographs.

That said, we agree with your suggestion to include “the relative concentrations of different compounds.” As you proposed in subsequent comments, we can address this by adding a “histogram comparison of all BPCA distributions.” (see below)

I really like to see a histogram comparison of all BPCA distributions in various samples mentioned in this study, including the experimental terrestrial samples, heated coal etc, as well as those previously reported in extraterrestrial samples such as Ryugu and Murchison etc.

Although B6CA/B5CA ratio and percentages are very informative, other smaller BPCAs may also be informative of structures. A histogram including all compounds will help illustrate the differences more thoroughly. I see data for Chang'E samples are listed in Table 1. But table listing is much less appealing than a histogram figure, and less easy for a visual comparison with other samples.

- Fig.3A and B are a bit confusing. I see two Y axis on the left and right sides of the plot (percent B6CA+B5CA, B6Ca/B5CA ration respectively), but it takes huge efforts to read the caption to figure out what is what. The figure needs to be more intuitive for comprehension.

Answer:

We appreciate the suggestion and agree that a histogram comparison of all BPCA distributions would be valuable. We have replaced the original Fig. 3A and B (see below) with new histograms, which are easier to interpret by audience. Please refer to the updated Fig. 3A and B (Line-146).

However, we believe it is important to retain Table 1 in the main text, as it provides the first quantitative results of polycyclic aromatics in lunar samples.

In addition, as aforementioned, there are currently no comparable data for extraterrestrial samples, which limits our ability to make direct comparisons.

(original Fig. 3A and 3B)

- Have there any published BPCA distributions of graphene sheet? In particular, if the 4-6 nm, 10 by 10 to 12 by 12 graphene sheet would generate similar BPCA distributions (e.g., B6CA to B5CA ratio), the data would be a welcome support for the paper. Line 155: Theoretically, a graphene sheet with a 10×10 benzene-ring array would give a B6CA/B5CA ratio of 2.18, and a 12×12 array would give a ratio of 2.73. This is theoretical. The actual BPCA distribution would depend on the efficiency of nitric acid oxidation (e.g, using the conditions reported in this paper). Has anyone conducted experiment to show this?

Answer:

Thanks for your kind concern. To our knowledge, no one has conducted this specific experiment. However, we expect nitric acid oxidation to efficiently convert graphene-like structures into BPCAs, as it has been demonstrated to effectively convert fullerenes and carbon nanotubes into BPCAs (Please refer to Line-328 to 330).

In response to your suggestion, we searched for graphene standards online and contacted several graphene manufacturers. Unfortunately, graphene standards or materials of similar

sizes (~4 nm) are not available. We have noted this limitation in the revised manuscript. Additionally, we have provided further information to support our estimation of the size of polycyclic aromatic structures in the lunar soil. Please see the corrections at Line-196 to 206.

- If the Mare plain region has a shorter impact history than other regions on the moon, the chances of completely eliminating other organic matter and transforming them into graphene would be lower. This contradicts with the points made in the paper somewhat.

It is very surprising that PYGCMS revealed no organics and PAHs at all. It is hard to imagine impacts have been so fully covered the whole region and left no original organics. Impacts cannot possibly eliminate all compounds, unless overlapping impacts have occurred to cover every inch of the lunar soil, at least in the sampling region? Some explanations are needed here.

Answer:

Meteorites, particularly carbonaceous chondrites, are known to be rich in organic matter. In this study, we propose a mechanism in which organic matter from meteorites is destroyed or converted into graphene-like materials under the high temperatures and pressures generated during impact events. This process explains why small organic molecules are rarely found on the Moon. In this scenario, the graphene-like materials are derived from the conversion of organic matter in meteorites, rather than from organic matter in the lunar soil itself. Consequently, the frequency of impacts is directly correlated with the abundance of graphene-like materials in lunar soils.

Another possible pathway involves methane implanted by solar winds. This methane, present in the lunar soil, may also be converted into graphene-like materials under the high temperatures and pressures of impact events. In this case as well, the more frequent the impacts, the greater the abundance of graphene-like materials in the lunar soils.

It is important to clarify that our proposed mechanism does not involve the destruction of organic matter or graphene-like materials in lunar soils by meteorite impacts. Instead, we suggest that impacts facilitate the conversion of organic matter (from meteorites or solar wind-implanted methane) into graphene-like materials. Therefore, the more frequent the impacts, the more graphene-like materials are produced, rather than eliminated.

The absence of an appreciable atmosphere on the Moon makes impact shocks significantly

more destructive compared to those on Earth. On Earth, organic matter such as amino acids, sugars, and monocarboxylic acids in meteorites often survives impacts. However, this is far less likely on the Moon due to the extreme conditions generated by impacts in the absence of atmospheric buffering.

Regarding the Py-GC-MS analysis, as discussed earlier, the graphene-like structures identified are non-volatile and refractory. Additionally, the heating process in Py-GC-MS is conducted under helium, not oxygen, ensuring that the analysis reflects pyrolysis (thermal decomposition) rather than combustion.

- Line 275, what is the concentration of distilled nitric acid?

Answer: It is 68%. We give this information in the revised manuscript (Line-354).

- Line 247. Asteroids (and carbonaceous chondrites falling onto Earth, like Murchison) also do not atmosphere and are subjected to impacts. They contain lots of soluble organic compounds and organics did not get converted into graphene.

Answer:

We acknowledge that the original sentence may have caused confusion. To clarify, we have modified the sentence into: “The search for organic carbon and biomolecules on other planetary bodies, such as Mars and early Earth, can be informed by our findings on the Moon.” (Line-25 and 321).

- Table 2, what is the bulk $\delta^{13}\text{C}$ value of the lunar soil?

Answer:

While this data would indeed be valuable for comparison, we were unable to perform the analysis due to insufficient sample availability.

- Line 280: “after addition of 100 μL water into the PTFE liner to prevent explosion of the ampoule”. What does this mean?

Answer: It is clarified. Please see Line-359: “To prevent ampoule explosions due to heating, 100 μL of water was added to the PTFE liner to balance the vapor pressure inside and outside

the ampoules.”

Reviewer #2 (Remarks to the Author):

Background Info

Zhong et al., have undertaken a comprehensive analysis of polycyclic aromatic hydrocarbons (PAHs) in the Chang'E 5 lunar soil samples for the first time. They detected highly condensed graphene sheet-like PAHs and demonstrated that they are unlike those found in terrestrial settings. Using isotopic evidence they indicate that the PAHs could have formed from bulk meteoritic organic matter (both soluble/free and insoluble/macromolecular organic matter). The authors then suggest that this may mean that smaller (soluble organic molecules), such as amino acids may thus have been destroyed during impact events on the surface of the moon. Additionally, the authors suggest that similar effects could lead to the destruction of small organic molecules on other planetary surfaces, with minimal atmospheres or no atmosphere at all.

Major Comments

Figure 3a and 3b – Are the error bars included for the kerogen data? If so, are they just smaller than the white data point symbol?

Figure 3a and 3b – The graphs are somewhat confusing what does the colour bar show and what does the white dot indicate? This needs to be described in the figure caption.

Answer:

Thanks for your kind concern. The errors for the kerogen data are less than 2% and are therefore not shown. We have clarified this in the revised manuscript (Line-150).

Following Reviewer #1's suggestion, we have replaced the original Figure 3a and 3b (see below) with a percentage bar chart of BPCA compositions (Line-146). This new figure is easier to interpret. The original Figure 3c has been moved to Figure 5 (Line-277).

(original Figure 3a, 3b and 3c)

Figure 3c – I suggest you to put data for insoluble organic matter isolated from meteorites. It is difficult to understand how similar or dissimilar the graphene-like PAHs you have reported are to the major component in carbonaceous chondrite meteorites.

Furthermore, it would be better to also discuss in text more clearly about the differences between the PAHs within the lunar soil samples and those in meteorites or the aromatic rich portions that compose the insoluble organic matter in meteorites. There is a study from quite some time ago that uses nitric acid to break down the insoluble organic matter within the

Murchison CM2 carbonaceous chondrite ([https://doi.org/10.1016/0016-7037\(77\)90076-X](https://doi.org/10.1016/0016-7037(77)90076-X)). I suggest you to look at it and see if there is any way to compare their data to yours. Another paper discusses the general structure of meteoritic insoluble organic matter and might be useful as well (doi: 10.1111/j.1945-5100.2010.01122.x).

Answer:

We appreciate the suggestion to include data on insoluble organic matter (IOM) isolated from meteorites. However, no such data are currently available for comparison. We also found the study you referenced, in which the Murchison meteorite was oxidized with nitric acid to produce BPCAs. To our knowledge, this is the only report on BPCA analysis of extraterrestrial samples.

That said, the BPCA method used in that study differs significantly from ours, making direct comparisons impractical. In our study, BPCAs are obtained through nitric acid oxidation of samples in sealed ampoules at 180°C (high temperature and pressure) for 8 hours, whereas the referenced study employed reflux with nitric acid for 27 hours. We discussed this in the revised manuscript (Line-164 to 168). Our BPCA method has been widely used for analyzing polycyclic aromatics in terrestrial samples (Line-67) and, to our knowledge, is the first to be applied to extraterrestrial samples.

Although direct comparisons based on BPCA data are not feasible, we have included a discussion in the revised manuscript (Line-168 to 181) on how the polycyclic aromatics in the Chang'e-5 lunar soils compare to those in carbonaceous chondrite meteorites, based on results from other analytical methods, including the one you kindly reminded of.

Table 1 – The caption states that a 25% uncertainty was used, why? Please explain this.

Answer:

It is clarified in the revised manuscript:

“The uncertainties in the abundance of polycyclic aromatics were calculated by applying a 25% error to the conversion factor (5.7) between total BPCAs and polycyclic aromatics (see Methods)”. (Line-101)

“The conversion factor represents an average value for all these chemicals, with an estimated uncertainty of 25%, corresponding to the relative standard deviation of conversion factors

derived from individual chemicals.” (Line-331)

Did you try to search for small organic molecules? From what I can understand you haven't searched for these molecules, but you suggest they might have been erased. I suggest you to look for amino acids or carboxylic acids, which are relatively abundant in meteorites. There are a number of methods that have been utilised recently to detect amino acids in the Ryugu samples for very small sample sizes (~1-2 mg to 10's or 100's mg of sample, see <https://doi.org/10.3390/life13071448>).

Answer:

Thank you for the suggestion. In response, we analyzed approximately 80 soluble organic compounds, which are commonly detected in extraterrestrial samples. Most of these compounds were undetectable in our study. This finding supports our conclusion that small organic molecules are erased and converted into refractory graphene-like structures during meteorite impacts. Additional information on this point can be found at Line-48 to 50, and Line-311 to 319.

Furthermore, we note that previous studies of lunar samples have also found little evidence of organic matter on the Moon, as discussed at Line-33 to 43.

Line 218-222 – The authors state that methane from solar wind can contribute to the solar regolith. It would be useful to explain whether a combination of solar wind derived methane and insoluble organic matter in meteorites or other carbon bearing phases in meteorites (e.g. carbonate) could generate the isotopic signatures of the graphene-like PAHs.

Answer:

To our knowledge, heating carbonates cannot produce graphene-like structures. However, heating any organic matter, including methane, at sufficiently high temperatures can yield graphene-like structures (Line-129 to 131). Previous studies have reported relatively positive $\delta^{13}\text{C}$ values for methane (+15‰ to +60‰) in lunar samples (Line-252).

It is important to note that insoluble organic matter (IOM) is also referred to as “kerogen-like” or “macromolecular organic matter”, and IOM is rich in polycyclic aromatic chemicals (Line-168 to 170). In our previous manuscript, we provided $\delta^{13}\text{C}$ values for meteoritic kerogen-like materials, which typically exhibit quite negative $\delta^{13}\text{C}$ values (-3‰ to -35‰). In the revised

manuscript, we have replaced the term “kerogen-like” with “insoluble organic matter (IOM)” for clarity.

We agree with your suggestion that a combination of lunar methane and IOM could generate the isotopic signatures of the graphene-like polycyclic aromatic hydrocarbons (PAHs). This idea has been incorporated into our discussions (Line-248 to 252).

Indeed, previous work simulating impacts has shown that small molecules in the interior portions of meteorites can survive impact events (e.g. DOI: 10.1089/ast.2008.0327).

Answer:

It is good to know this reference. The study simulated the high pressures associated with meteorite impacts (up to 30 GPa), but the temperature used in the experiments was relatively low (58°C), as noted in the paper.

Furthermore, it has also been shown that ¹³C and D can be preferentially released from insoluble organic matter in meteorites when simulating impact events. This is because the portions of the insoluble organic matter that contain these heavy isotopes are likely bound by weaker bonds to the structure of the insoluble organic matter (see DOI: 10.1111/j.1945-5100.2007.tb00238.x). Therefore, it is possible that some isotopic fractionation between the meteoritic organic matter and the end product would have occurred. As such, I think you need to show that the lunar soils are devoid of small organic molecules like amino acids and carboxylic acids to give weight to your proposed theory of formation for the graphene-like PAHs.

Answer:

Thank you for bringing this reference to our attention. We have included discussions on isotopic fractionation in the revised manuscript. According to this reference, the $\delta^{13}\text{C}$ of insoluble organic matter (IOM) becomes more depleted when “weak bond structures” are lost during impacts. This result aligns with the findings of the paper we cited, which showed more depleted $\delta^{13}\text{C}$ in IOM residues after Py-GC-MS analysis.

The reference you suggested supports our conclusion that the graphene-like material in the Chang'E-5 lunar soils is not merely a remnant of IOM, as it would otherwise exhibit more depleted $\delta^{13}\text{C}$ values than the IOM of meteorites. This implies the presence of another non-

aromatic source contributing positive $\delta^{13}\text{C}$ values. We have incorporated this reference into the revised manuscript (Line-239).

Overall, the article, once revised, could be appropriate for publication in Nature Communications, but it would require additional analysis. The authors should utilise sensitive techniques to search for soluble organic compounds, such as those commonly found in meteorites (e.g. amino acids and carboxylic acids). This will enable them to tell if the graphene-like PAHs formed from bulk meteorite organic matter or from another source, such as the interaction of insoluble meteoritic organic matter and solar wind CH_4 that was implanted into the lunar surface.

Minor Comments

Please change all instances of polycyclic aromatics to polycyclic aromatic hydrocarbons (including in the title), as this is the correct term and also polycyclic aromatics isn't strictly grammatically correct.

Answer:

Thank you for your suggestion. The term "polycyclic aromatic hydrocarbons (PAHs)" typically refers to smaller molecules with more hydrogen atoms, which does not accurately describe the graphene-like structures we identified in the lunar soils. Similarly, the term "polycyclic aromatic compounds" is not entirely suitable, as the chemicals we are discussing lack specific molecular structures.

The BPCA method we employed targets aromatic structures spanning a wide range, from small PAHs to fullerenes, carbon nanotubes, and graphene sheets. Therefore, the term "polycyclic aromatics" may better align with the chemical structures targeted by our BPCA method. This term is also commonly found in studies published in ACS journals, such as:

- Ball M, Zhong Y, Wu Y, et al. Contorted polycyclic aromatics[J]. Accounts of Chemical Research, 2015, 48(2): 267-276.
- Wegner H A, Reisch H, Rauch K, et al. Oligoindenopyrenes: a new class of polycyclic aromatics[J]. The Journal of Organic Chemistry, 2006, 71(24): 9080-9087.

- Goldfinger M B, Crawford K B, Swager T M. Directed electrophilic cyclizations: efficient methodology for the synthesis of fused polycyclic aromatics[J]. Journal of the American Chemical Society, 1997, 119(20): 4578-4593.

The English should be improved throughout, as there are many cases where words are missing (e.g. the) or used incorrectly (e.g. is instead of are). While I was able to follow the article for the most part, there were a number of sentences that I had to re-read several times and if the overall English was improved, I think it would greatly aid in the communication of the authors scientific findings.

Answer:

Thank you for the suggestion. We had our manuscript language-polished by a professional editing service.

Dear Reviewers,

We would like to extend our sincere gratitude for your time and effort in reviewing our manuscript and providing valuable feedback.

In the attached "Answers to Reviewer Comments" document:

- Black text indicates the reviewers' comments.
- Blue text represents the authors' responses to the reviewers' comments.
- Yellow highlighting in the track-changed manuscript marks the revisions made.
- The notes labeled "Line-x" refer to the line numbers in the revised manuscript, corresponding to the specific corrections and responses to the comments.

We deeply appreciate your constructive suggestions, which have greatly strengthened our manuscript. Thank you once again for your support.

Faithfully yours,

Gan Zhang

Professor of Organic Geochemistry

Guangzhou Institute of Geochemistry

Chinese Academy of Sciences

Answers to Reviewer Comments

Reviewer #1 (Remarks to the Author):

The manuscript has been improved in the first round of revision. However, I would like to see following issues more carefully addressed:

Line 18-19: Please provide a clear explanation why the particular isotope range indicates de novo formation.

Answer:

We have revised the text as suggested.

Original text:

"The $\delta^{13}\text{C}$ values of the polycyclic aromatics range from $-5.0 \pm 0.6\%$ to $+3.6 \pm 1.3\%$, suggesting a de novo formation mechanism during carbonaceous meteorite impacts, involving the conversion of non-aromatic organic matter into polycyclic aromatics."

Revised text:

"Meteorite impacts are believed to be the most probable events responsible for the presence of polycyclic aromatics in lunar soils. However, the $\delta^{13}\text{C}$ values of the polycyclic aromatics in Chang'E-5 lunar soil range from $-5.0 \pm 0.6\%$ to $+3.6 \pm 1.3\%$, which are significantly more enriched than those found in carbonaceous meteorites. This enrichment suggests a *de novo* formation mechanism during carbonaceous meteorite impacts, involving the conversion of non-aromatic organic matter—which is more enriched in $\delta^{13}\text{C}$ —into polycyclic aromatics." (Lines 18–24)

Fig.1 legend: The model structure shown is a polycyclic aromatic hydrocarbon or PAH. But the paper kept using "polycyclic aromatics" throughout the paper. If the purpose is to include aromatic structures containing oxygen, nitrogen heteroatoms etc., this should be clarified. How would the presence of oxygen or nitrogen affect the nitric acid

oxidation products?

Answer:

The fused benzene ring structures characterized and quantified by the BPCA method can either represent the entire molecule or just a part of the molecule. The terms “polycyclic aromatic hydrocarbons,” “polycyclic aromatic substances,” or “polycyclic aromatic chemicals” cannot fully describe the latter case. There can be other structures in the molecules, such as hydroxyl, azyl and heterocycle etc, which are not targeted by BPCA method.

In the revised manuscript, we have clearly defined the term “polycyclic aromatics” and explained why the more common terms mentioned above are not used (Lines 66–70).

Additionally, we have revised the legend of Fig. 1 for clarity. The original text stated: *“During oxidation, a single fused benzene ring (highlighted in red) is preserved and substituted with carboxylic groups derived from adjacent rings or side chains.”*

This has been revised to replace “rings or side chains” with “carbon atoms” (Lines 74–76). Theoretically, if the atom adjacent to a single fused benzene ring is a heteroatom, it will not result in BPCAs, as the carboxylic group of BPCA must be derived from a carbon atom adjacent to the fused benzene ring.

Fig.2 legend: Key HPLC parameters, like mobile phase and HPLC column condition should better be stated in the figure caption.

Answer:

Thanks for the suggestion. Key HPLC parameters are included in the Fig.2 legend in the revised manuscript. Please see L93-95.

Line 106: High temperature processes under vacuum and absence of oxygen. If oxygen is present, the organics will be converted into CO₂ in high temperature. Also, clarify how high the temperature needs to be to produce PAHs.

Answer:

We have clarified this point in the revised manuscript (L112-115). The high-temperature processes do not necessarily require a vacuum or the absence of oxygen. For example, fossil fuel and biomass combustion occur in the presence of air, where the primary combustion product is CO₂, but a small amount of particles enriched in polycyclic aromatics is also released. We acknowledge that the original statement, “the primary sources of polycyclic aromatics include biomass burning, fossil fuel combustion,” was unclear and may have caused confusion. This has been revised to: “On Earth, the primary sources of polycyclic aromatics include incomplete combustion of biomass and fossil fuels.”

To the best of our knowledge, no study has explicitly determined the specific temperature threshold required for polycyclic aromatics formation. However, we provide an example: heating wood chips at 200°C for 5 hours is sufficient to produce polycyclic aromatics.

Fig.3 legend. Again, people often say polycyclic aromatic hydrocarbons or PAHs. What is the reason of saying polycyclic aromatics in this paper? If there is a special reason for using polycyclic aromatics rather than PAHs, the reason should be explained clearly.

Answer:

Please see a same answer above to this question.

Line 240: Need more clear reasoning here why and what isotope fractionation and processes are involved, so that the carbon isotope values of B5CA and B6CA indicate de novo formation processes. For example, what percentage mixing of IOM and soluble compounds with different isotopic values would lead to observed values.

Fig.4 legend: Apart from mixing different isotope pools, what other isotope fractionation processes may be involved in the de novo synthesis?

Answer:

Thanks for the suggestion. We add one paragraph focusing on discussions of the related

isotopic fractionation. Please see L266-279.

line 252: For the observed carbon isotope values of B5CA and B6CA, try to make some estimates of different sources: how much of the solar wind implanted methane with $\delta^{13}\text{C}$ of +15 to 60 permil methane may have contributed to the formation of lunar organics? what isotope fractionation processes may be involved?

Answer:

Thank you for this valuable suggestion. Although addressing this issue is challenging due to limited data, we have attempted to provide meaningful discussions on the topic. Please refer to the added paragraph discussing this point in detail (Lines 280–296).

Line 272: What are the possible isotope effect of these processes?

Answer:

The fragmentation should not result in isotopic fractionation, as it does not involve carbon loss or transformation into non-polycyclic aromatic structures (L317-318).

Line 302: Decomposition and evaporation?

Answer:

Thank you for pointing this out. The correct term should indeed be “evaporation.” In this context, we discuss the abundance of elemental carbon (C) rather than the abundance of organic matter in lunar soils. In the case of decomposition, carbon may remain in the lunar soil rather than being lost from it.

Reviewer #2 (Remarks to the Author):

Thank you very much for revising your manuscript. After the revisions that you have made I believe there is sufficient evidence to support your conclusions that small organic molecules within impacting meteorites could be destroyed and form the highly aromatic graphene sheet-like organic matter present in the lunar soil samples returned by the Chang E 5 mission. Therefore, I am happy for the article to be accepted in Nature

communications.

Answer:

We sincerely appreciate your positive feedback and support for our manuscript. Thank you for recognizing the significance of our findings, and we are delighted to hear that you recommend acceptance in *Nature Communications*.